# 4EHP and NELF-E regulate physiological ATF4 induction and proteostasis in disease models of *Drosophila*

Kristoffer Walsh[1], Hidetaka Katow[1], Hannah Junn[1], Deepika Vasudevan[1,3], Christoph Dieterich[2] & Hyung Don Ryoo[1] ✉

Cells adapt to proteostatic and metabolic stresses, in part, through stress activated eIF2α kinases that stimulate the translation of ATF4. Stress-induced ATF4 translation is regulated through elements at ATF4 mRNA's 5' leader. In addition to eIF2α kinases, ATF4 induction requires other regulators that remain poorly understood. Here, we report an ATF4 regulatory network consisting of eIF4E-Homologous Protein (*4EHP*), NELF-E, the 40S ribosome, and eIF3 subunits. Specifically, we found that the mRNA cap-binding protein, *4EHP*, was required for ATF4 signaling in the *Drosophila* larval fat body and in disease models associated with abnormal ATF4 signaling. *NELF-E* mRNA, encoding a regulator of pol II-mediated transcription, was identified as a top interactor of 4EHP in a TRIBE (Targets of RNA Binding through Editing) screen. Quantitative proteomics analysis revealed that the knockdown of *NELF-E* or *4EHP* commonly reduced several subunits of the 40S ribosome (RpS) and the eIF3 translation initiation factor. Moreover, reduction of *NELF-E, 4EHP, RpS12, eIF3l*, or *eIF3h* suppressed the expression of ATF4 and its target genes. These results uncover a previously unrecognized ATF4 regulatory network consisting of 4EHP and NELF-E that impacts proteostasis during normal development and in disease models.

Cells adapt to external or physiological stress in part by inducing the expression of stress-responsive genes. Among known adaptive mechanisms is the Integrated Stress Response (ISR), a signaling pathway initiated by stress-activated eIF2α kinases[1,2]. ATF4 (Activating Transcription Factor 4) is a major transcription factor that mediates ISR, inducing the transcription of various genes involved in proteostasis and amino acid biosynthesis[3–8]. Reflecting the broad role of ISR in responding to diverse cellular stress, there is an increasing list of metabolic and degenerative diseases associated with abnormal ISR signaling.

Multiple eIF2α kinases responding to diverse conditions of stress can initiate ISR signaling. For example, GCN2 gains activity in response to amino acid deprivation, and PERK is best characterized for its activation by endoplasmic reticulum (ER) stress. In mammals, the eIF2α kinase HRI is activated in response to mitochondrial stress[9–11]. In *Drosophila*, certain types of mitochondrial stress also activate PERK, as in the case of *parkin* mutants that impair mitophagy[12–14]. Notably, loss of *parkin* underlies rare forms of familial Parkinson's Disease in humans[15], suggestive of a possible role of ISR in this disease.

The immediate consequence of eIF2α phosphorylation is to transform it into an inhibitor of the guanine exchange factor eIF2B, thereby suppressing the delivery of the ternary complex (TC, consisting of eIF2-GTP-Met-tRNA$_i^{Met}$) to the 40S ribosome[16]. The net effect is a general attenuation of mRNA translation initiation. The mRNAs of metazoan *ATF4* and yeast *GCN4* evade such suppression, and in fact, undergo preferential translational induction to further mediate ISR's

[1]Department of Cell Biology, NYU Grossman School of Medicine, New York, USA. [2]Department of Internal Medicine III, University Hospital Heidelberg, Heidelberg, Germany. [3]Present address: Department of Cell Biology, University of Pittsburgh School of Medicine, Pittsburgh, USA. ✉e-mail: hyungdon.ryoo@nyulangone.org

gene expression program. Such induction of ATF4 and GCN4 relies on the properties of their unique 5′ leaders that contain regulatory upstream Open Reading Frames (uORFs)[17–20]. uORFs are present in numerous transcripts, and they often reduce the translation of downstream ORFs because of translation termination after uORF translation. However, the 40S ribosomes on the mRNAs of *ATF4* and *GCN4* can continue to scan and reinitiate translation downstream of uORFs[20–22]. Re-initiation after termination is possible if the 40S can retain essential translation initiation factors such as eIF3[23–25]. During the translation of standard ORFs, re-initiation doesn't occur in part because the 40S-eIF3 interaction becomes unstable and eIF3 is eventually lost[24]. But on short uORFs, such as those found in *ATF4* or *GCN4*, some eIF3 are still retained by the 40S after uORF translation, allowing the 40S-eIF3 complex to recruit a fresh TC and other initiation factors for re-initiation at the downstream main ORF[26–29]. Supporting this notion, studies have found that the reduction of certain eIF3 subunits can impair uORF-mediated translational regulation of yeast *GCN4*, plant *AtbZip11*, and *ATF4* in human cell lines[23–25,30,31].

Here, we report a previously unrecognized pathway that regulates 40S ribosome subunits, eIF3, and ATF4 expression. In the *Drosophila* larval fat body with physiological stress and ATF4 signaling, we found that the depletion of *4EHP* or *NELF-E* reduced ATF4 and its target gene expression. *Drosophila parkin* (*park*) mutants dramatically stimulated ATF4 signaling, which was also suppressed by *4EHP* loss. Consistently, *4EHP* affected ATF4-associated pathological phenotypes, including *parkin* mutants' lethality and light-dependent retinal degeneration. We profiled mRNAs that bind to 4EHP's cap-binding domain using TRIBE (Targets of RNA Binding through Editing)[32], and among the top interactors was *NELF-E* mRNA. 4EHP was necessary for *NELF-E* expression, and *NELF-E* knockdown reduced ATF4 levels. Among the genes commonly reduced after the depletion of *NELF-E* or *4EHP* were components of the 40S ribosome (*RpS* genes) and a subunit of the eIF3 translation initiation factor complex. Reduction of *RpS12, eIF3l*, or *eIF3h* suppressed physiological ATF4 signaling in the *Drosophila* fat body without affecting the expression of control transgenes. Together, these findings uncover an ATF4 regulatory network consisting of *4EHP*, *NELF-E*, 40S ribosome, and eIF3 subunits, impacting proteostasis during normal development and in disease models.

## Results

### A screen identifies *4EHP* as a gene required for physiological ATF4 expression in the fat body

The *Drosophila* genome encodes a single *ATF4* ortholog annotated as *cryptocephal* (*crc*)[33]. A well-established transcriptional target of *Drosophila* crc is *Thor*, an ortholog of *4E-BP1*[34–36]. The single intron of *Thor* contains crc/ATF4-binding sites, and this regulatory sequence was previously fused to the coding sequence of *DsRed* fluorescent protein to generate an ATF4 reporter known as *Thor^intron^-DsRed* (also referred to as *4E-BP^intron^-DsRed*)[34] (Fig. 1a). During the late third instar larval stage, *Thor^intron^-DsRed* expression is induced in the larval fat body (Fig. 1b, c), a metabolic tissue analogous to the mammalian liver and the adipose tissue. We had previously shown that the *Thor^intron^-DsRed* signal in the fat body is dependent on GCN2 and ATF4, suggestive of physiological amino acid deprivation stress during normal development[34].

We performed a small-scale RNAi screen for their capacity to regulate crc/ATF4 signaling, utilizing the Gal4/UAS method[37] to express RNAi transgenes with the fat body specific *dcg-Gal4* (Fig. 1b). The 184 RNAi lines targeted either known translation regulators or the *Drosophila* homologs of ribosome-associated proteins[38]. The RNAi lines that reduced the *Thor^intron^-DsRed* signal comparable to that caused by *crc/ATF4* RNAi were marked as hits. Our screen identified 19 hits, and the RNAi line targeting *4EHP* (*eIF4E-Homologous Protein*, also known as *eIF4E2*) was among those to cause the starkest decreases in DsRed fluorescence (Fig. 1c–j and Supplementary Data 1). The

knockdown of *4EHP* in the fat body caused a slight delay in development (Supplementary Fig. S1), and to assess any general effect on protein synthesis, we also examined a control transgene with a single ORF, *UAS-GFP* expression driven by the *dcg-Gal4* driver. Neither *crc* nor *4EHP* RNAi reduced this control GFP expression in the fat body, supporting their specific effects on crc/ATF4 signaling (Fig. 1f′–h′, j). To validate the role of *4EHP*, we employed a hypomorphic loss-of-function *4EHP* mutant allele, *4EHP^CP53^*, with significantly reduced *4EHP* expression[39]. The *Thor^intron^-DsRed* signal was also reduced in this *4EHP^CP53^* background (Supplementary Fig. S2). Going beyond the analysis of the *Thor* reporter, we validated that *Thor* transcripts were significantly reduced in *4EHP* RNAi and *4EHP^CP53^* larvae, as measured through RT-qPCR (Fig. 1k).

4EHP is homologous to eIF4E, with a conserved domain that binds the 7-methylguanosine cap (m7GpppN, where N is a nucleotide) of mRNAs. Although eIF4E is required for a bulk of cap-dependent mRNA translation, 4EHP has a weaker cap-binding affinity and associates with other proteins, such as 4E-T, to bind to a more specific set of target mRNAs[40–43]. Perhaps because 4EHP works with other such factors, overexpression of *4EHP* alone was insufficient to induce *Thor^intron^-DsRed*, and RNAi lines that target *4E-T* reduced this reporter in the larval fat body (Supplementary Fig. S3). We decided to further characterize 4EHP due to its possible specificity in gene expression control.

We assessed *Drosophila* crc protein levels in the larval fat body through western blots. Overexpression of *crc* (specifically the *RB* transcript) in this tissue using the *dcg-Gal4* driver generates a clear anti-crc band near the 25 kDa marker (Fig. 1l, lane 3). Control larval fat bodies also show a crc protein band, albeit weaker than that caused by the transgene overexpression (Fig. 1l, lane 1). Knockdown of *4EHP* significantly reduced the crc band intensity (Fig. 1l, m), indicating that *4EHP* is required for physiological crc/ATF4 expression in the larval fat body.

### Loss of *4EHP* impacts gene expression related to metabolism and translation

To better understand 4EHP's role in gene expression, we performed RNA-seq analysis of the larval fat body transcripts with or without *4EHP* RNAi. With the criterion of $p$-adjusted <0.05, 183 transcripts were significantly downregulated and 218 upregulated after *4EHP* RNAi (genotype: *dcg-Gal4/UAS-4EHP* RNAi) as compared to controls (genotype: *dcg-Gal4/+*) (Fig. 2a and Supplementary Data 2). The Principal Component Analysis (PCA) plot showed a clear separation of the two genotypes (Fig. 2b). *crc* transcript levels did not change significantly by *4EHP* RNAi (Fig. 2a), supporting the idea that crc/ATF4 signaling is impacted through a post-transcriptional mechanism.

As transcriptional changes in stressed cells do not necessarily lead to similar changes in protein levels, we also performed a quantitative proteomics analysis in control and *4EHP* RNAi larval fat body (Fig. 2c and Supplementary Data 3). With the criterion of $p$-adjusted < 0.05, 364 proteins were detected at significantly reduced levels and 333 at higher levels in *4EHP* RNAi fat body samples.

Interestingly, 12 of the top 13 enriched GO Terms for those proteins significantly reduced by *4EHP* RNAi were related to various metabolic processes (Fig. 2d). To assess changes in metabolism, we compared the steady-state metabolic profiles of homozygous *4EHP^CP53^* larvae with those from control *w^1118^* samples (3 independent samples for each genotype). Among 147 common cellular metabolites assessed, 107 of them were detected from at least three samples, and 89 were detected from all six samples (Supplementary Data 4). Nine metabolites were significantly lower in the *4EHP^CP53^* larvae (Supplementary Fig. S4a and Supplementary Data 4). We note that seven of these nine downregulated metabolites are either amino acids or their metabolic products. Six of these seven (excluding L-alanine) were metabolites in either the serine-one-carbon pathway or the urea cycle (Supplementary Fig. S4b, c). Changes in these metabolites correlate with the

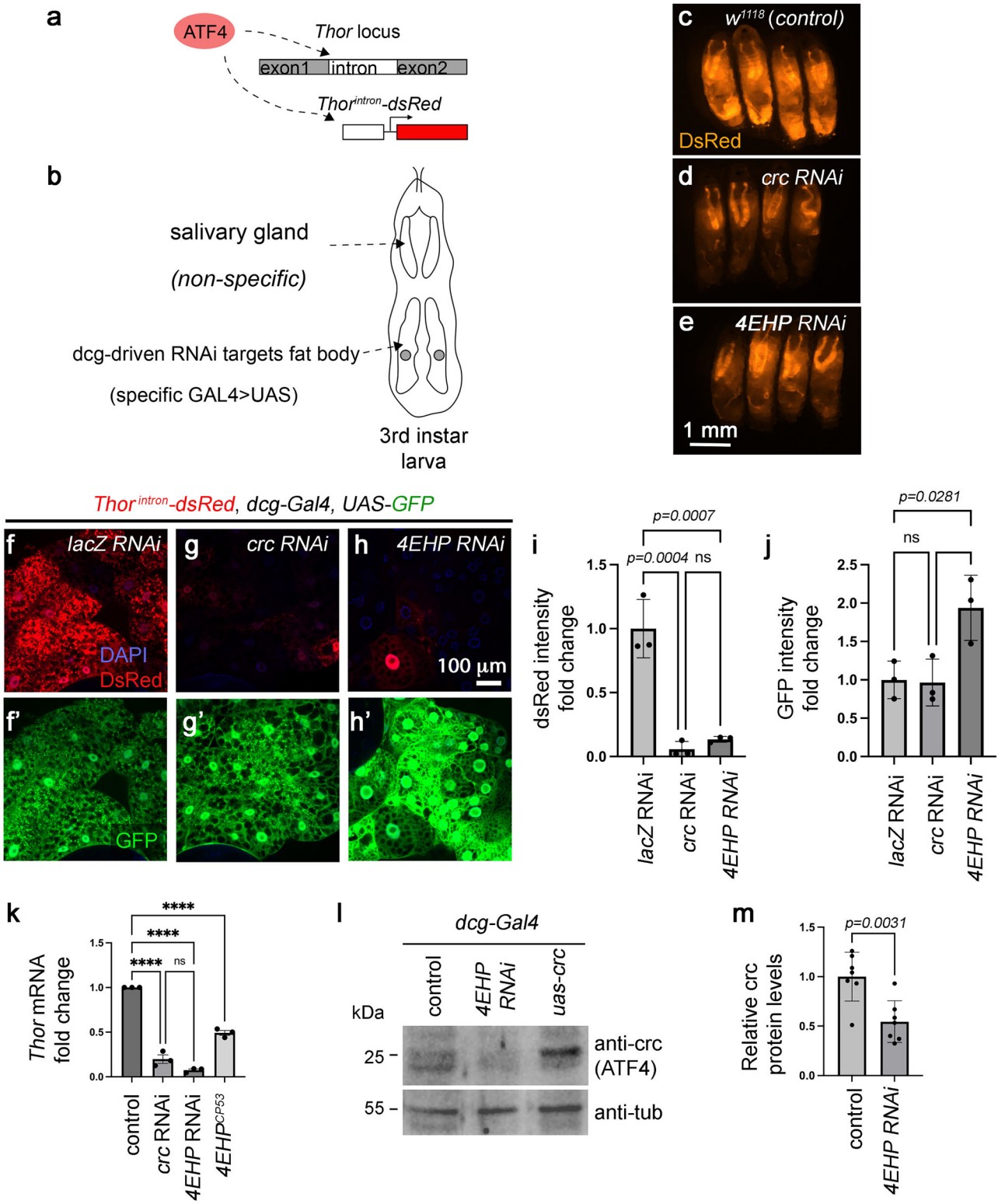

reduction of serine-one-carbon pathway enzyme transcripts, including *aay*, *Nmdmc*, *Gnmt* (Fig. 2a, e) in the *4EHP* RNAi dataset. *aay*, which encodes an L-phosphoserine phosphatase that mediates serine biosynthesis, was also found significantly reduced in our quantitative proteomics data (Fig. 2c, e).

*4EHP^{CP53}* homozygous females had a slightly shorter lifespan compared to control females, but the mutant male flies had lifespans similar to controls when reared under standard conditions of nutrients (Supplementary Fig. S5). We suspected that the amino acids available

in standard food were sufficient to sustain the viability of *4EHP^{CP53}* flies beyond 80 days despite indications of decreased crc/ATF4 signaling. To assess the effect of nutritional deprivation, we isolated day-1 adult flies and subjected them to total nutrient starvation. Under these conditions, wild-type control adults retained over 50% of their population for 60 h, followed by a drop in population by 72 h, and a total population loss occurred by 96 h. Under equivalent conditions, *4EHP^{CP53}* homozygotes and *4EHP* RNAi adults died at a significantly faster rate (Fig. 2f, g). To test if protein deficiency contributes to a

**Fig. 1 | *4EHP* is required for the expression of the crc (ATF4) and its target, *Thor intron-DsRed*.** **a** A schematic diagram of the *Thor intron-DsRed* reporter. The *Thor* intron, containing ATF4 binding sites, drives DsRed expression. **b** A diagram of the RNAi screen. The fat body-specific *dcg-Gal4* was used to drive double-stranded *RNA* expression in the larval fat body for RNAi knockdown of targets. The *Thor intron-DsRed* reporter is expressed both in the salivary gland and the fat body, but only the fat body signal is affected by the knockdown of crc/ATF4 signaling mediators. **c**–**e** Representative whole larvae images expressing the *Thor intron-DsRed* reporter. Similar fluorescence patterns were consistently observed across multiple independent larvae (here shown as $n = 4$ per genotype). **c** Negative control larvae crossed to $w^{1118}$ instead of an RNAi transgene. **d, e** RNAi against *crc* (ATF4) (**d**), or *4EHP* (VDRC #38399) (**e**) suppressed DsRed fluorescence in the region with fat body tissues, but not the non-specific signal from the anterior salivary glands. (**f**–**h**) *Thor intron-DsRed* signals (red) from the dissected third instar larval fat body. A negative control with *lacZ* RNAi (**f**), and the equivalent flies crossed to *crc* RNAi (**g**) or *4EHP* RNAi (VDRC #38399) (**h**). **f'**–**h'** Control *dcg-Gal4 > UAS-GFP* expression (green) assessed in the indicated genotypes. Quantification of the DsRed intensities (**i**) and GFP intensities (**j**) indicates that the effect of *4EHP* RNAi is specific for the *Thor intron-DsRed* and not for the control *dcg-Gal4 > UAS-GFP* expression. Data in (**i**) and (**j**) represent results from three biological replicates ($n = 3$). Statistical significance was assessed through ordinary one-way ANOVA followed by Tukey's multiple comparisons test. ns indicates not significant. **k** *Thor mRNA* fold change, as assessed through RT-qPCR from the indicated genotypes. Data represent results from three biological replicates (each averaged from technical triplicate, $n = 3$). Statistical significance was assessed through ordinary one-way ANOVA followed by Tukey's multiple comparisons test. **** indicates $p < 0.0001$. **l** Anti-crc (top gel) and anti-tubulin western blots (bottom gel) from third instar larval fat body extracts. The control sample (lane 1) shows a moderate-intensity anti-crc band that disappears in *4EHP* RNAi (lane 2) or becomes more intense after *crc* overexpression (lane 3). (**m**) Quantification of the normalized crc band intensity. Data represent results from seven biological replicates ($n = 7$) per genotype. Two-tailed Welch's *t* test was used for statistical analysis. All bar graphs (**i**, **j**, **k**, **m**) show mean values +/− SD (Standard Deviation).

shorter lifespan in these flies, we assessed survival after reintroducing protein (1.6% BSA) into their diet. This protein-only diet made *4EHP^CP53^* and *4EHP* RNAi flies less vulnerable to starvation (Fig. 2i, j). In particular, *4EHP^CP53^* flies displayed a survival curve similar to control flies when reared with 1.6% BSA (Fig. 2i). Equivalent knockdown of *crc* in the fat body also made the flies vulnerable to total starvation (Fig. 2h), and adding back 1.6% BSA to the diet abolished the survival difference with control flies (Fig. 2k). Together, these results indicate that the loss of *4EHP* reduces the levels of certain amino acids and makes flies vulnerable to starvation.

One of the top 13 enriched GO terms in the proteins reduced by *4EHP* RNAi was "translation" (Fig. 2d). Among the established initiation factors, eIF4A, eIF3l, and eIF3k were detected significantly lower in *4EHP* RNAi samples (Fig. 2c and Supplementary Data 3). Not all initiation factors were reduced by *4EHP* RNAi. For example, three subunits of the eIF2 complex were detected at significantly higher levels in samples with *4EHP* knockdown (Fig. 2c). Western blots validated the increase in total eIF2α caused by *4EHP* knockdown, but there wasn't a concomitant increase in phospho- eIF2α. As a result, the ratio of phospho-eIF2α to total eIF2α decreased in *4EHP* knockdown samples (Supplementary Fig. S6a–c). The ribosome subunits also showed an interesting pattern: All of the significantly reduced ribosomal proteins in the *4EHP* RNAi proteome dataset were subunits of the 40S ribosome (referred to as RpS proteins), while many 60S subunits (RpL proteins) were detected at significantly higher levels (Fig. 2c and Supplementary Fig. S6d). While many RpS proteins were reduced, the transcript of RpS7 was the only 40S subunit component reduced in the RNA-seq dataset (Fig. 2a). Our observation is consistent with published reports that the decrease in one RpS subunit causes the reduction of other RpS subunit proteins while increasing RpL subunits and other ribosome biogenesis factors[44,45].

## *4EHP* loss affects proteostasis phenotypes

Since ATF4 signaling is deregulated in many diseases, we examined 4EHP's possible effect in *Drosophila* disease models. *Parkin* (*park*) loss-of-function variants underlie rare forms of Parkinson's Disease, and *park* encodes a protein that helps to clear defective mitochondria[46,47]. Previous studies had reported that *Drosophila park* loss of function mutants activate PERK and promote eIF2α phosphorylation[12,48]. To examine the effect of *park* loss, we used *park^25^* and *park^D21^* alleles, which are null alleles with deletions in the coding sequence[49,50]. As reported previously, *park^25^/park^D21^* trans-heterozygotes survived to adulthood as long as the larvae were reared under uncrowded conditions (20 or fewer larvae in a vial). We found that *park^25^/park^D21^* adult flies showed intense *Thor intron-DsRed* reporter expression indicative of strong crc (ATF4) signaling (Fig. 3a, b).

To assess the role of *4EHP* in this disease model, we crossed *4EHP^CP53^* into the *park^25^/park^D21^* background (genotype: *Thor intron-DsRed/+; park^25^, 4EHP^CP53^/park^D21^, 4EHP^CP53^*). These flies expressed significantly reduced *Thor intron-DsRed* signals compared to flies with *park^25^/park^D21^* alone (Fig. 3a, b). As noted by others, a large fraction of the *park^25^/park^D21^* adult flies died within the first day after eclosion (Fig. 3c). By contrast, introducing *4EHP^CP53^* in this background (genotype: *Thor intron-DsRed/+; park^25^, 4EHP^CP53^/park^D21^, 4EHP^CP53^*) significantly suppressed the lethality of *park^25^/park^D21^* flies (Fig. 3c).

To determine if crc/ATF4 signaling also affects *parkin* mutants, we employed the *crc^1^* loss-of-function allele. *crc^1^* homozygotes exhibit early developmental lethality, but we found that the *crc^1^/+* heterozygotes moderately but significantly ($p < 0.0293$) prolonged the lifespan of *park^25^/park^A21^* adults (Supplementary Fig. S7). These results indicate that the loss of *4EHP* suppresses crc/ATF4 signaling in *parkin* mutant flies, and excessive crc/ATF4 signaling contributes to the enhanced lethality of *parkin* mutants.

In another model associated with ATF4 signaling, we examined light-dependent retinal degeneration, a phenotype accelerated by the loss of *PERK* or *crc*[51]. To assess the role of *4EHP* in this model, we examined pseudopupils, which are trapezoidal patterns that appear in response to blue light deep in the retina. The pseudopupil forms only when ommatidial clusters maintain regular arrays of aligned photoreceptors and serves as a convenient tool to assess ommatidial integrity. As previously reported[51], *crc^GFSTF^* hypomorphs rapidly lost pseudopupils when reared under light (Fig. 3d). *4EHP^CP53^* flies also lost pseudopupils earlier than the control ($w^{1118}$) flies when reared under light (Fig. 3d). Such loss of pseudopupils did not occur when *4EHP^CP53^* mutant flies were reared in the dark (Fig. 3d). These results further support the idea that *4EHP* regulates crc/ATF4 signaling in disease models.

## 4EHP binds to *NELF-E* mRNA and *NELF-E* is required for ATF4 signaling

To gain mechanistic insights into the regulation of crc/ATF4 signaling by 4EHP, we searched for mRNAs that bind to 4EHP's cap-binding domain. We specifically used an approach referred to as Targets of RNA-binding proteins Identified By Editing (TRIBE), involving an Adenine Deaminase Acting on RNA (ADAR) fused to an RNA-binding protein of interest[32,52]. ADAR edits adenine to inosine on the bound mRNAs, and those edited RNAs can be identified through RNA-seq. We generated *UAS*-transgenes with ADAR fused to wild-type 4EHP. As a negative control, we fused ADAR to the 4EHP^W114A^ mutant, which impairs the mRNA 5'cap-binding pocket[39] (Fig. 4a). We drove the expression of these constructs in the fat body using *dcgGal4* and analyzed polyA-containing RNAs from third instar larval fat bodies for RNA editing profiles. The mRNAs of *crc* or any eIF2α kinases did not

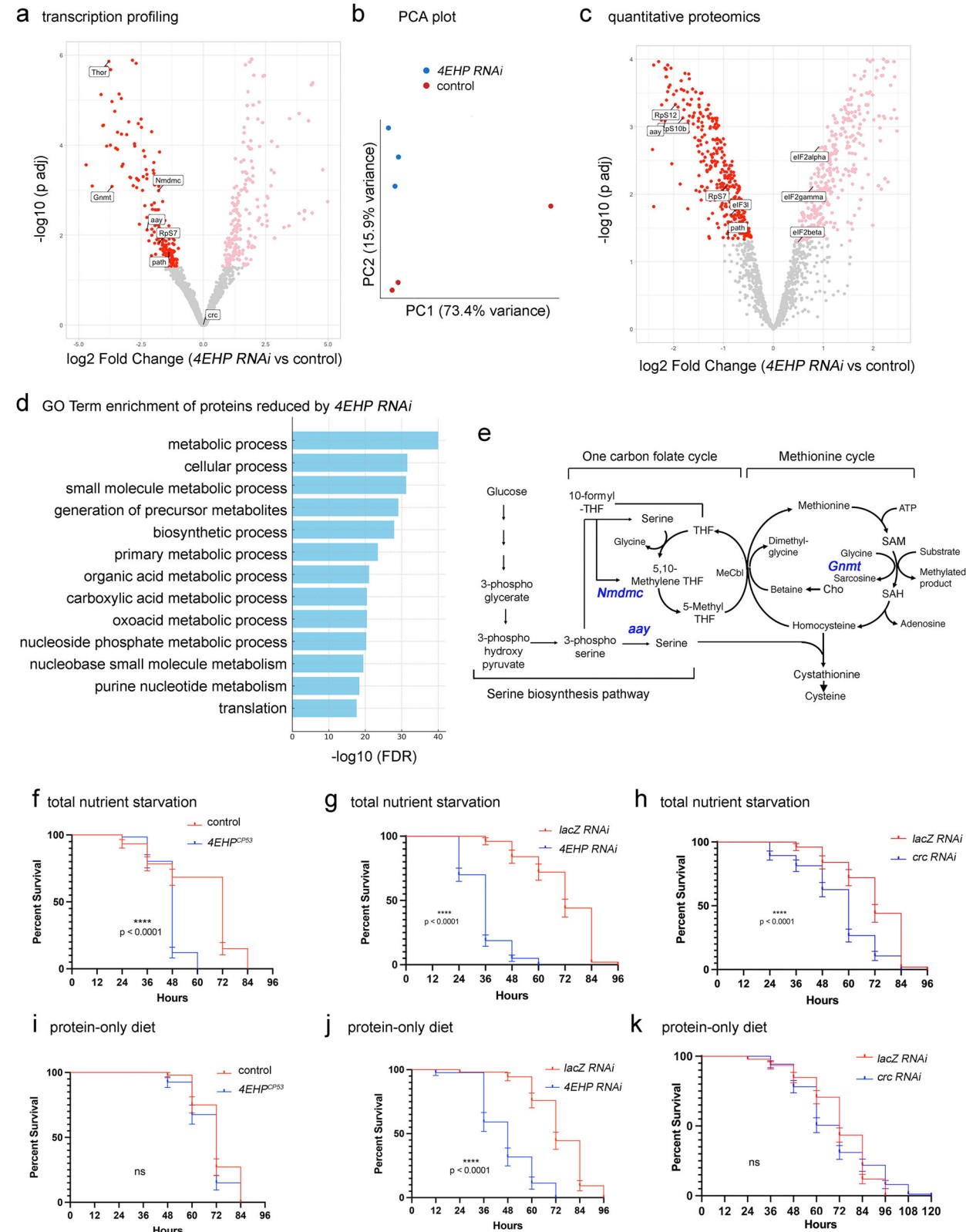

score as significantly edited targets (Fig. 4b, Supplementary Data 5). The preferentially edited mRNAs (binding scores > 10) were enriched with diverse GO Terms including cellular developmental process and cell differentiation (Fig. 4c). The mRNA most significantly edited by 4EHP-ADAR was *CG18132*, encoding an uncharacterized protein disulfide isomerase, predicted to assist in protein folding within the endoplasmic reticulum. The second highest was the mRNA of *Negative*

*Elongation Factor E* (*NELF-E*) (Fig. 4b, d). We considered *NELF-E* a high-confidence hit, as another A to I edit 30 bases downstream of the first had the 7th highest score (Fig. 4b, d). An mRNA encoding another subunit of the NELF complex, *NELF-A*, was also edited with significance: one site was ranked at 141, and a second *NELF-A* site was ranked at 1157 (Fig. 4b and Supplementary Data 5). *NELF-E* transcript levels were comparable between control and *4EHP* RNAi conditions (Fig. 4e), but

**Fig. 2 | Loss of *4EHP* reduces gene expression related to metabolism and translation. a** A volcano plot of gene expression changes caused by *4EHP* RNAi (VDRC #38939) in larval fat body tissue samples across three biological replicates per genotype (*n* = 3). Labeled are those genes involved in serine-one-carbon pathway (*Nmdmc, Gnmt, aay*), mRNA translation (*Thor, RpS7*), and amino acid transport (*path*). Differential gene expression was computed using DESeq2 with its default two-sided Wald test on a negative-binomial generalized linear model; *p*-values were adjusted with the Benjamini–Hochberg method. **b** A PCA plot of three samples for each genotype. Those flies crossed to *w1118* instead of *4EHP* RNAi were used as controls. **c** A volcano plot of the proteome changes caused by *4EHP* RNAi in the larval fat body. Labeled are those involved in amino acid metabolism or transport (*aay, path*) and mRNA translation (RpS12, RpS10b, RpS7, eIF3l, eIF2alpha, eIF2beta, eIF2gamma). Statistical significance was determined using a two-sided moderated *t* test (limma) with FDR correction (fdrtool) as implemented in the DEP2 R package. **d** Enriched GO Terms of the proteins reduced by *4EHP* RNAi. **e** A schematic diagram of the serine-one-carbon pathway with the transcripts reduced by *4EHP* RNAi in

blue. **f**–**k** Kaplan–Meier survival curves showing estimated survival probability ± standard error (SE) of flies subjected to starvation. One-day-old adults were subjected to the following conditions of nutrient restrictions. **f** Control (*w1118*, *n* = 82) and *4EHPCP53* homozygotes (*n* = 76) reared with no nutrients. **g** Flies with either *lacZ* (control, *n* = 50) or *4EHP* knocked down (*n* = 80) with the fat body driver (*dcg-Gal4*) reared without any nutrients. **h** Those with either *lacZ* or *crc* RNAi (*n* = 75) reared with no nutrients. **i** Comparison of control (*w1118*, *n* = 69) and *4EHPCP53* homozygotes (*n* = 55) reared with only 1.6% BSA. Note these two strains showed a difference in survival under total nutrient starvation (**f**), which disappears with the addition of 1.6% BSA in the diet. **j** Comparison of control (*lacZ* RNAi, *n* = 92) and *4EHP* RNAi (VDRC #38939) flies (*n* = 44) when reared with only 1.6% BSA in food. The difference in survival is less than that observed with total nutrient starvation (compare with **e**). **k** The equivalent experiment comparing *lacZ* and *crc* RNAi (*n* = 87). Data are presented for (**f**–**k**) with error bars reflecting SE (Standard Error). Log-rank analysis (two-sided) was used to assess statistical significance. *p*-values are indicated. ns = not significant.

our anti-NELF-E western blots revealed that *4EHP* RNAi reduced NELF-E protein levels (Fig. 4f, g). These results indicate that 4EHP binds to *NELF-E* mRNA to regulate its expression at a post-transcriptional level.

The NELF complex is composed of four subunits, and overexpression of *NELF-E* alone was neither sufficient to enhance *Thorintron-DsRed* reporter expression nor rescue the *ThorintronDsRed* levels in *4EHP* RNAi larvae (Supplementary Fig. S8). Late third instar larvae failed to emerge when *NELF-E* was knocked down with the *dcg-Gal4* at 25 °C, but we obtained viable larvae at 20 °C, albeit with a developmental delay (Supplementary Fig. S8e). Under these conditions, two independent RNAi lines targeting *NELF-E* reduced the *Thorintron-DsRed* reporter signal in the fat body but not the control *dcg-Gal4 > UAS-GFP* expression (Fig. 5a–d). We also observed that *NELF-E* RNAi reduced crc protein levels (Fig. 5e, f), corroborating our observations with *Thorintron-DsRed*. Consistently, flies with *NELF-E* RNAi died significantly faster than control *lacZ* RNAi under total starvation (Fig. 5g). Adding 1.6% of BSA as a protein source to the vial abolished such a difference between *NELF-E* RNAi and *lacZ* RNAi (Fig. 5h). These results indicated that, similar to *4EHP, NELF-E* is required for physiological crc/ATF4 signaling and survival under nutrient deprivation.

### *NELF-E* regulates metabolic pathways and RpS subunits
To further gain insights into *NEFL-E*'s role, we performed RNA-seq on *NELF-E* RNAi fat body tissue samples with *lacZ* RNAi samples as a control. We found that 312 transcripts were significantly induced, while 890 transcripts were found at significantly reduced levels in *NELF-E* RNAi samples (Fig. 6a and Supplementary Data 6; *p*-adjusted < 0.05). The PCA plot showed a clear separation between the *NELF-E* RNAi and control samples (Fig. 6b). Similar to what we had seen with *4EHP* RNAi, the three serine biosynthetic enzymes (*aay, CG6287*, and *CG11899*) and the onecarbon pathway enzyme *Nmdmc* were detected at significantly reduced levels in *NELF-E* RNAi fat body samples (Fig. 6a). These enzymes are established targets of ATF4 in *Drosophila* and mammals[5,48,53,54].

In addition to RNA-seq, we subjected these fat bodies to quantitative proteomics analysis (Fig. 6c and Supplementary Data 7). Those peptides that were reduced in *NELF-E* RNAi fat bodies, as compared to *lacZ* RNAi controls, were enriched with GO Terms that included serine transport and L-serine biosynthetic process (Fig. 6d). Consistently, the proteomics dataset from *NELF-E* RNAi samples showed reduced levels of L-serine biosynthetic enzymes such as *aay* and *CG11899* (Fig. 6c).

Previous studies had documented gene expression changes in cells deficient in *NELF*[55,56]. Because those studies had not made associations between NELF and ATF4, we re-examined one of the published datasets reported with *NELF-B* knockout mouse embryonic stem cells[56]. The transcripts specifically reduced in the knockout cells include the major enzymes that mediate the serine-one-carbon pathway, homologous to the *Drosophila* enzymes reduced by *NELF-E* RNAi

(Supplementary Fig. S9). Taken together, the gene expression profiling results support the idea that NELF regulates ATF4 target genes across phyla.

We note that mouse *NELF-B* knockouts had significantly reduced *ATF4* transcript levels[56]. However, analogous changes in the *Drosophila crc* expression did not occur in *Drosophila NELF-E* knockdown samples (Fig. 6a). This suggested that *Drosophila* NELF-E may utilize a posttranscriptional mechanism to regulate *crc* expression. To gain insight, we examined gene expression changes that were common between *4EHP* and *NELF-E* knockdown samples. There was an overlap between the transcripts impacted by RNAi of *4EHP* and *NEFL-E* (Supplementary Fig. S10a), with 83% of the transcripts significantly reduced in *4EHP* RNAi samples also significantly reduced after *NELF-E* RNAi. Similarly, the proteomics data showed that 63% of the proteins reduced in *4EHP* RNAi fat body were also reduced in *NELF-E* RNAi fat bodies (Supplementary Fig. S10a). The proteins commonly reduced in *4EHP* and *NELF-E* RNAi samples were enriched with GO terms such as metabolic processes and cytoplasmic translation (Supplementary Fig. S10b). Since crc/ATF4 expression is regulated at the level of mRNA translation, we further examined if translation initiation factors changed their levels in *4EHP* or *NELF-E* RNAi samples. eIF3l was the only eIF protein that was significantly reduced in both conditions. In addition, many 40S ribosome subunits (RpS proteins) were detected at significantly reduced levels (Supplementary Fig. S10a). The most strongly reduced peptides mapping to the RpS subunits were RpS10b, Rps27A, RpS20, and Rps12 (Fig. 6e). According to the published human 40S ribosome structure (PDB 7r4x), these subunits cluster together in a specific part of the ribosome (Fig. 6f). Therefore, we speculate that the reduction of *RpS12* transcripts in *NELF-E* RNAi fat body leads to the instability of the other subunit peptides.

The effect of *NELF-E* RNAi appeared selective as none of the RpL subunits were significantly reduced in the *NELF-E* RNAi or *4EHP* RNAi data (Fig. 6e and Supplementary Fig. S6). In fact, many RpL subunit peptides were detected at higher levels, possibly through a feedback regulatory mechanism against RpS reduction (Fig. 6e). Also detected at higher levels were the three eIF2 subunits, and the increase in the total eIF2α was validated through western blots (Supplementary Fig. S11). Phospho-eIF2α levels didn't increase together with total eIF2α, resulting in a reduced ratio of phospho-eIF2α to total eIF2α in the *NELF-E* RNAi samples (Supplementary Fig. S11).

To test if the reduction of RpS subunit contributes to ATF4 signaling impairment, we employed the *RpS12S2783*, a loss of function allele caused by a transposable P-element insertion. The homozygotes were lethal as expected, but the *RpS12S2783*/+ heterozygotes were viable and reduced *Thorintron-DsRed* signals in the larval fat body as compared to controls (Fig. 6g). Similar to the case of *4EHP* and *NELF-E* knockdown, *RpS12S2783*/+ did not reduce the expression of control *dcg-Gal4 > UAS-GFP* (Fig. 6g, h), indicating that crc/

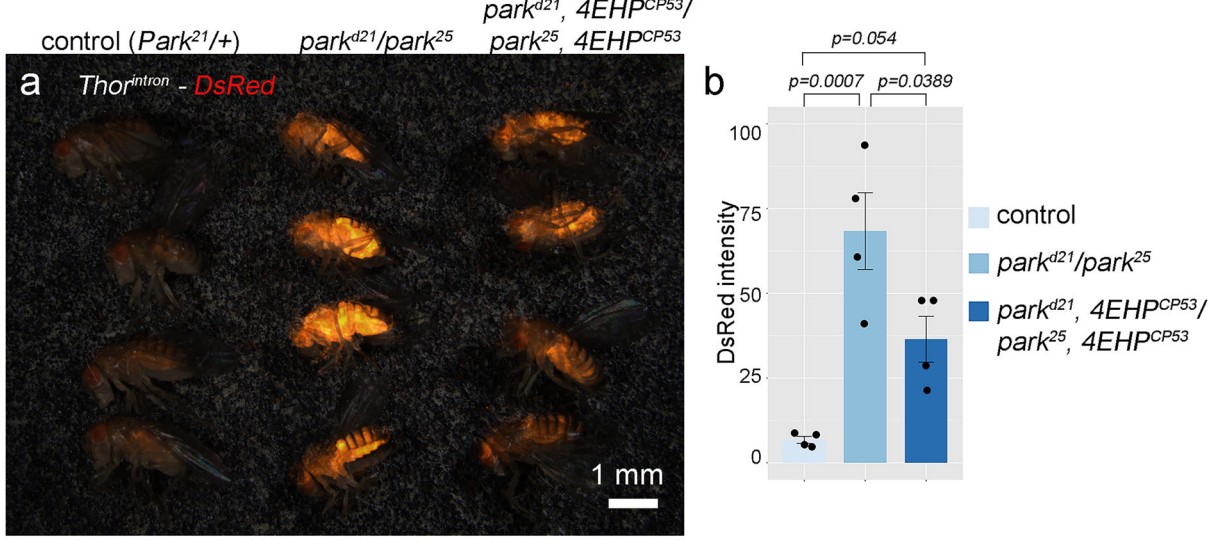

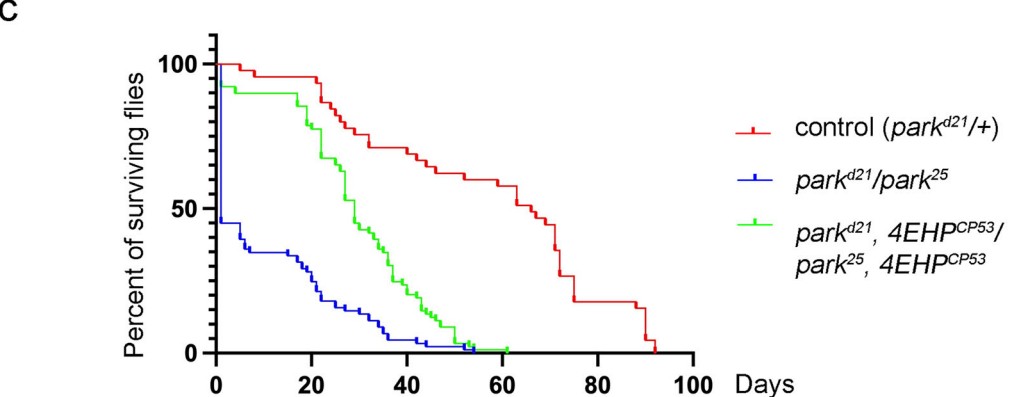

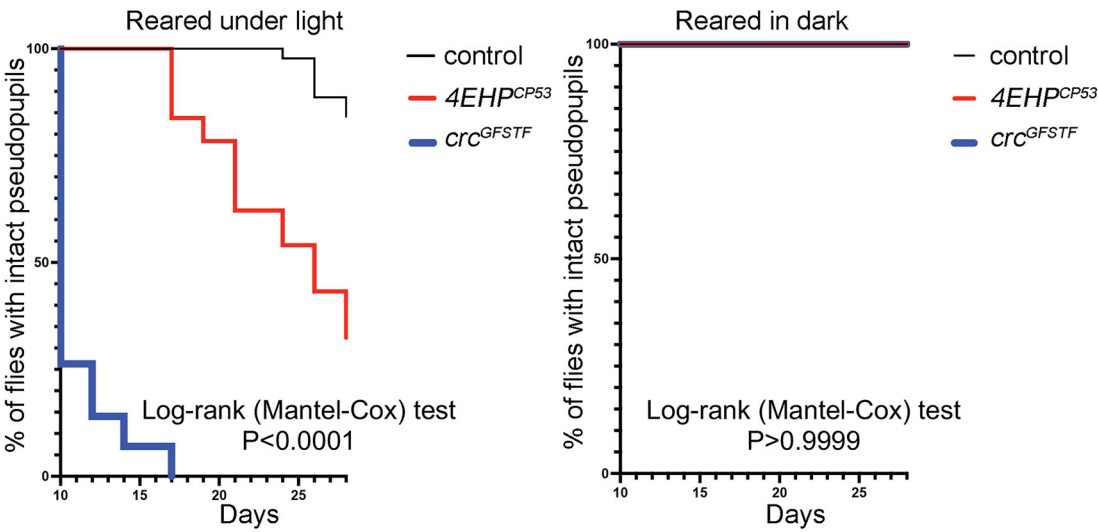

**d** ommatidial integrity assessed through pseudopupils

ATF4 signaling is specifically sensitive to the reduction of the *RpS12* gene dosage.

## crc/ATF4 signaling is selectively impaired by the reduction of eIF3 subunits

We next turned our attention to the eIF3 complex as eIF3l was commonly reduced by the knockdown of *4EHP* and *NELF-E*, and because eIF3 partners with the post-termination 40S subunit for re-initiation downstream of uORFs[23,24,30]. The knockdown of certain subunits caused larval lethality or severe developmental delay, but fat body-specific expression of RNAi lines targeting eIF3 subunits, *eIF3h* and *eIF3l*, and the eIF3-associated factor *eIF3j*, did not interfere with larval development, with most flies successfully reaching pupal stages (Supplementary Fig. S12). We found that *eIF3l* knockdown in the fat

**Fig. 3 | The loss of *4EHP* affects the outcome of proteostasis phenotypes in degenerative disease models. a–c** The *parkin* (*park*) loss-of-function phenotype is suppressed by the loss of *4EHP*. **a** *Thor^intron^-DsRed* expression in one-day-old adult flies of the indicated genotypes. *park^D21^/park^25^* flies have intense DsRed induction, which is partially suppressed in the *4EHP^CP53^* homozygous background. **b** Quantification of the DsRed signal from flies shown in (**a**). One-way ANOVA and Tukey's HSD were used to assess statistical significance. **c** The lifespan of the flies of the indicated genotypes. *park^D21^/park^25^* flies show high levels of lethality immediately after eclosion, and very few flies survive more than 40 days. *4EHP^CP53^* in that genetic background significantly enhanced survival. Log-rank (two-sided) was used

to assess statistical significance. $p < 0.0001$ between control and *park^D21^/park^25^*. $p < 0.0001$ between *park^D21^/park^25^* and *park^D21^, 4EHP^CP53^/park^25^, 4EHP^CP53^*. **d** Light-dependent retinal degeneration is accelerated in *4EHP^CP53^* flies. Pseudopupils were used to assess retinal integrity in live flies. (Left) A graph showing the percentage of flies with intact pseudopupils when reared under light. *4EHP^CP53^* flies (red line) exhibit accelerated retinal degeneration as compared to control (*w^1118^*) flies (black line). *crc^GFSTF^* flies (blue line) have early onset retinal degeneration. (Right) Flies with intact pseudopupils when reared in the dark. The log-rank test (two-sided) was used to assess statistical significance between all three genotypes in light conditions (left, $p < 0.0001$) and in dark conditions (right, $p > 0.9999$).

body reduced *Thor^intron^-DsRed* signals in the larval fat body as compared to controls (Fig. 7a–d).

To independently validate the role of eIF3, we knocked down *eIF3h*. Of note, our previous screen had scored *eIF3h* RNAi as a suppressor of the crc/ATF4-signaling reporter expression[22]. Two independent RNAi lines targeting *eIF3h* (VDRC 106189 and BDSC 55603) suppressed *Thor^intron^-DsRed* expression when expressed in the larval fat body (Fig. 7e–g and Supplementary Fig. S13). These knockdown conditions did not block a control *GFP* transgene expression (Fig. 7e', f', h), further supporting a specific effect of *eIF3h* knockdown on ATF4 signaling. We also examined an allele of *eif3h, k09003*, with a transposable element (P-element) inserted in its coding sequence. Homozygotes *eif3h^k09003^* did not survive to late larval stages, but the heterozygote 3^rd^ instar larvae had weaker *Thor^intron^-DsRed* expression in the fat body (Supplementary Fig. S14).

To gain further insight into *eIF3h* RNAi's effect, we generated a UAS line with the *crc* 5' leader preceding the DsRed reporter (Fig. 7i). This transgene will not capture potential crc (ATF4) regulation that acts through the protein-coding sequence but is designed to report regulatory inputs at the 5' leader upstream of the main ORF. We first tested the stress-inducible nature of this reporter in eye imaginal discs. When the reporter was expressed with the eyespecific *GMR-Gal4*, no DsRed expression was detected, indicating that the *crc* 5' leader inhibited the main ORF translation in unstressed cells. Expressing *Rh1^G69D^*, a missense allele of *Rh1* that induces crc/ATF4 when expressed in eye imaginal discs[57], robustly induced DsRed expression (Supplementary Fig. S15). The equivalent experiment in the *PERK* −/− background abolished DsRed induction, indicating that *crc 5' leader-DsRed* reports eIF2α kinase-mediated crc induction (Supplementary Fig. S15). When driven with the fat body-specific Gal4 driver, we detected clear *crc 5' leader-DsRed* signals that reported physiological ISR activity (Fig. 7j, k). Such reporter signal was significantly suppressed when *eIF3h* was knocked down (Fig. 7l, m). *eIF3h* RNAi did not significantly reduce phospho-eIF2α levels (Supplementary Fig. S16). These results support the idea that depletion of *4EHP* or *NELF-E* reduces eIF3 levels, and physiological ATF4 signaling is suppressed by the reduction of eIF3 subunits.

While our data demonstrates the requirement of *eIF3h*, overexpression of *eIF3h* alone was not sufficient to induce *Thor^intron^-DsRed* (Supplementary Fig. S17a–c), perhaps because other eIF3 subunits could be limiting. Consistently, overexpressing *eIF3h* did not rescue the suppression of *Thor^intron^-DsRed* caused by *NELF-E* RNAi (Supplementary Fig. S17d, e).

Finally, we examined if the newly identified regulators are themselves under the control of crc/ATF4 signaling. We found that the transcript levels of *4EHP*, *NELF-E*, and *eIF3h* did not change significantly in late larval fat bodies as examined through RT-qPCR (Supplementary Fig. S18). These results do not support a feedback regulatory relationship between crc/ATF4 and *4EHP*, *NELF-E*, *eIF3h*.

## Discussion

In this study, we report that *4EHP* and *NELF-E* form a regulatory axis required for physiological crc/ATF4 signaling. 4EHP uses its 5' cap-binding domain to bind *NELF-E* mRNA and promote *NELF-E* expression.

*4EHP* and *NELF-E* regulate a highly overlapping set of genes and proteins, including subunits of the 40S ribosome and the eIF3 complex. Reducing these components, such as *Rps12* or *eIF3h*, suppresses crc/ATF4 signaling. While the *Drosophila* lines used in this study were not backcrossed to a common parental strain, key results were corroborated with independent RNAi lines or with classical mutant alleles. These results support a previously unrecognized relationship between *4EHP* and *NELF-E* and their roles in crc/ATF4 signaling (Fig. 8).

Compared to the mRNA cap-binding protein eIF4E, there has been only a limited understanding of 4EHP function. While both proteins can bind to mRNA caps, only eIF4E can recruit eIF4G for translation initiation. 4EHP reportedly represses the translation of many mRNAs upon binding[39,42]. A small number of in vivo studies had identified 4EHP's role in *Drosophila* development and in mouse behavior[39,58,59]. We find that *Drosophila* 4EHP has a specific role in regulating NELF-E and ATF4 signaling. ATF4 is an established regulator of nonessential amino acid biosynthesis, and the fat body is an organ that readily reprograms metabolism in response to amino acid deprivation[60–62]. Consistently, the loss *4EHP* reduced specific aspects of amino acid metabolism.

Our results show that the loss of *4EHP* impacts the proteostasis-related phenotypes of two degenerative disease models in *Drosophila*. One of the degenerative disease models we examined was the *parkin* mutant, which reportedly has higher PERK activity[12]. Consistently, we found that *parkin* loss strongly activated PERK's downstream ATF4 signaling. Moreover, our data suggest that the excessive ATF4 signaling associated with *parkin* loss contributes to the phenotype.

Our TRIBE screen provides insights into the molecular function of 4EHP. The overall results indicate that 4EHP binds to a small fraction of cellular mRNAs. The results are consistent with the finding that 4EHP has a significantly weaker 5' cap binding affinity compared to eIF4E[41]. 4EHP may bind to specific target mRNAs through a combination of its capbinding domain function and its interaction with other RNA-binding proteins. Among the highest-scoring 4EHP interactors was the mRNA of *NELF-E*. Because NELF-E protein levels were found to be reduced after *4EHP* RNAi, we conclude that 4EHP stimulates NELF-E expression after binding to the mRNA. Whether 4EHP acts as a translational regulator of *NELF-E* is unclear and beyond the scope of this study. It is possible that 4EHP stimulates *NELF-E* translation, analogous to its role in translating certain mRNAs[63,64]. Alternatively, 4EHP may affect *NELF-E* expression indirectly, perhaps by regulating mRNA maturation or transport.

NELF was initially characterized as a complex that promotes RNA polymerase II pausing at promoter proximal sites of certain genes[65,66]. More recent studies have reported NELF's alternative role in regulating the cap-binding complex on nascent mRNAs[67–69]. The net effect of *NELF* depletion is a change in gene expression, with a reduction in transcripts involved in heat shock, ERK signaling, and innate immune response[55,56,70–72]. Our study shows that *NELF-E* is required for ATF4 signaling. Specifically, we found a clear reduction of crc/ATF4 in the larval fat body after *NELF-E* knockdown. Consistently, gene expression profiling data showed that *NELF-E* RNAi in the fat body caused a strong reduction in established ATF4 target gene expression.

**a** 4EHP-ADAR

**b** mRNAs preferentially edited by wt 4EHP-ADAR

**c** GO Term Enrichment

**d**

**e**

**f** NELF-E western blot

**g**

While the association between NELF and ATF4 target genes had evaded notice in previous studies, our own analysis of publicly available datasets further supports the relationship between NELF and amino acid biosynthesis. For example, a microarray study of *NELF* depleted *Drosophila* S2 cells reported a reduction in two out of three enzymes that convert 3-phospho-glycerate to Serine (*aay* and *CG11899*)[55]. Moreover, a study of mouse embryonic stem cells reported that *NELF-B* knockout caused the reduction of two out of three serine biosynthetic enzymes (*Phgdh* and *PSAT1*) and three well-established one-carbon pathway enzymes (*Shmt2, Mthfd1, Mthfd2*)[56]. These results

suggest a relationship between NELF and ATF4-induced amino acid metabolism that is largely conserved across cell types and between species.

We did not see a change in *Drosophila crc* transcript levels after *NELF-E* knockdown, suggesting that *Drosophila NELF-E* regulates *crc* through a post-transcriptional regulatory mechanism. Among the proteins reduced in *NELF-E* RNAi fat body were multiple subunits of the 40S ribosome and an eIF3 subunit. This was interesting because eIF3 works together with the 40S ribosome during translation initiation[73]. Many other translation initiation factors dissociate from the ribosome

**Fig. 4 | 4EHP binds to *NELF-E* mRNA, dependent on its cap-binding domain, to regulate *NELF-E* expression. a** The design of the TRIBE experiment to identify 4EHP-binding mRNAs. The RNA editing enzyme, ADAR, was fused to either the wild-type 4EHP or to the cap binding-deficient 4EHP $^{WII4A}$ mutant. These chimera-proteins are designed to edit the mRNAs (from A to I) that they bind. **b** A plot of mRNAs preferentially edited by 4EHP-ADAR. The binding score on the y axis shows the log-likelihood of mRNA editing by 4EHP wild type-ADAR on a specific nucleotide divided by that from the equivalent 4EHP $^{WII4A}$ mutant. Highlighted are two different edited sites on *NELF-E* and *NELF-A*. **c** The enriched GO Terms of the 4EHP targets with binding scores above 10. **d** A schematic diagram of the *NELF-E* mRNA and the relative positions of the sites edited by 4EHP wild type-ADAR. Gray shows

the UTRs, and black indicates the coding sequence. **e** *NELF-E* transcript levels in control and *4EHP* RNAi RNA-seq dataset with three biological replicates per genotype ($n = 3$). A two-sided $t$ test was used to determine significance. Data presented are mean values +/− SE. **f** Anti-NELF-E (top gel) and anti-tubulin (bottom gel) western blots from fat bodies of control ($w^{III8}$), or with *4EHP* (VDRC #38399) or *NELF-E* RNAi (VDRC #21009). **g** Quantification of relative NELF-E protein band intensities as compared to that of the control sample across three biological replicates (each averaged from technical duplicate, $n = 3$). Data presented are mean values +/− SD. An ordinary one-way ANOVA followed by Tukey's multiple comparisons test was used to assess statistical significance. *p*-values are indicated. ns = not significant.

once uORF translation initiates, but eIF3 remains bound to the translating 80S ribosome[26,27,29]. Such properties make eIF3 a prime candidate factor required for re-initiation after uORF translation, because eIF3 retained by the ribosome may help recruit other essential translation initiation factors that were lost during uORF translation (Fig. 8). Consistent with this idea, studies in yeast, *Arabidopsis*, and with human cells reported that eIF3 is required for the translation of ORFs that contain regulatory uORFs[23,24,30,31]. We found that the knockdown of *eIF3h* impairs ATF4 signaling, but not the expression of a control transgene. eIF3h is one of the non-core subunits of eIF3, with no homologs in *S. cerevisiae*. The data indicate that crc/ATF4 signaling is particularly sensitive to a reduction in *eIF3h* levels. In our experiments, a moderate reduction of 40S subunits or eIF3 did not affect the expression of *UAS-GFP*, a control transgene with a single ORF, and these results imply that ribosomes and eIF3 are not rate-limiting for that transcript expression. On the other hand, we found that the uORF-containing *crc* was highly sensitive to even a moderate reduction of *RpS12*, *eIF3l*, or *eIF3h*. This could reflect the fact that eIF3 is gradually lost from the ribosome during uORF translation, and there is a limiting amount of 40S-eIF3 downstream of the uORF to help re-initiate translation (Fig. 8).

In summary, our results support an ATF4-regulatory axis involving *4EHP*, *NELF-E*, 40S subunits and eIF3, which are required for physiological and pathological ATF4 signaling activity in *Drosophila*. The study provides insight into an ATF4 regulatory mechanism with pathological implications.

## Methods

### Fly strains

All flies were reared in a standard cornmeal-agar diet supplemented with molasses unless otherwise specified. Due to sexual differences in metabolism and ATF4 signaling, males were analyzed unless otherwise stated. All gene overexpression and RNAi experiments were done using the Gal4/UAS binary expression system[37]. *NELF-E* RNAi crosses were maintained at 20 °C due to larval lethality at higher temperatures. All other crosses were maintained at 25 °C. The following fly stocks were used for this study: $w^{III8}$ (BDSC # 5905), *Thor* (*4E-BP*)$^{intron}$- *DsRed*[54], *dcg-Gal4*, *UAS-crc*[33], *UAS-lacZ* RNAi[74], *UAS-ATF4* (*crc*) RNAi (VDRC #109014), *UAS-4EHP* RNAi (VDRC #38399 and BDSC #36876), *UAS-NELF-E* RNAi (VDRC #21009, BDSC #32835), *UAS-eIF3h* RNAi (VDRC #106189, BDSC #55603), *UAS-eIF3l* RNAi (VDRC #107267), *UAS-4E-T* RNAi (VDRC #101047 and # 34755), *4EHP*$^{CPS3\ 39}$, *park*$^{D21\ 50}$, *park*$^{25\ 49}$, *Perk*$^{e01744}$ (BDSC #85557).

### Molecular cloning

To generate flies expressing 4EHP-ADAR constructs for TRIBE, fusion protein sequences were subcloned into the pUAST attB vector. The 4EHP-RD CDS (full sequence Flybase ID FBtr0303160) composed the N-terminus of the fusion protein, followed by a 3' linker joined to the 5' end of the ADAR-RN catalytic domain sequence. In accordance with previous reports using TRIBE, the ADAR sequence contains a mutation at E488Q to "hyper"-activate ADAR's function to improve sensitivity[52]. The ADAR catalytic domain was followed by a second linker, joined by

a C-terminal V5 tag. For 4EHP$^{WII4A}$-ADAR, the TGG codon for tryptophan at nucleotides 340–342 was substituted for GCT to encode alanine. The full construct was generated via Invitrogen GeneArt Gene Synthesis Services. EcoR1 and Kpn1 sites were introduced to the construct via High Fidelity PCR amplification using the following primers: ADAR-F-EcoR1 5′- GCGGAATTCATGAGCATGGAGAAAGTAGC-3′; ADAR-R-Kpn1 5′- AGTGGTACCTCACGTAGAATCGAGACCGA-3′. The resulting product was verified to be free of PCR-induced mutations using sanger sequencing with services provided by Genewiz. The pUAST attB plasmids containing the verified complete construct were injected into the line 24482 by BestGene Inc to insert into the 2$^{nd}$ chromosome at a 51 C locus.

To generate the *crc 5'UTR-DsRed* reporter, the *crc* RA isoform was used, and its *ATF4* main ORF was replaced with the DsRed coding sequence. The resulting *crc* 5'UTR fused to DsRed was subcloned into the pUAST attB vector and injected into the VK37 strain to insert into the 2$^{nd}$ chromosome at 22A3.

### Metabolic profiling

Late larval stage samples from control ($w^{III8}$) and *4EHP*$^{CPS3}$ homozygous flies were analyzed by the NYU Metabolomics Core Resource Laboratory using the hybrid LCSM assay. Triplicate samples for each genotype were processed after scaling the metabolite extraction to a measured aliquot (5 larvae/mL). A panel of 147 metabolites were assessed, and 89 metabolites were detected in all 6 samples after background threshold correction. To assess differential metabolite expression, metabolite peak intensities were extracted to a library of m/z values and retention times developed with authentic standards. Intensities were extracted with an in-house script with a 10 ppm tolerance for the theoretical m/z of each metabolite, and a maximum of 30 s retention time window. A cocktail of isotope-labeled amino acid standards was spiked into the metabolite extraction solvent cocktail. Peak intensities were extracted according to a library of m/z values and retention times for the doubly labeled (13 C and 15 N uniform) amino acids.

### Nutrient starvation experiments

For full nutrient starvation experiments, day-1 adult male flies were collected and relocated to vials containing medium composed of 1.5% agarose in PBS, which were incubated at room temperature. At twelve-hour intervals, deaths were recorded. The experiment was run until all populations perished from starvation. For protein-only diet experiments, a medium composed of 1.5% agarose and 1.6% BSA in PBS was used.

### Photoreceptor degeneration assay

We collected 0-1 day AE flies from each genotype (Light: $w^{III8}$, $n = 44$. *4EHP*$^{CPS3}$, $n = 37$. *crc*$^{GFSTF}$, $n = 57$. Dark: $w^{III8}$, $n = 37$. *4EHP*$^{CPS3}$, $n = 15$. *crc*$^{GFSTF}$, $n = 38$.) and kept in regular cornmeal vials covered by parafilm made holes for a gas exchange. These vials were put into two cardboard boxes (11.5 cm × 11.5 cm × 14 cm) which are with or without a lid. The boxes are put into a 25 °C incubator during the assay. For the light source, we put an LED light pad (B4 Tracing Light Box with Internal Cord + Foldable Stand, 14.2 * 10.6 Inches Light Board for Tracing,

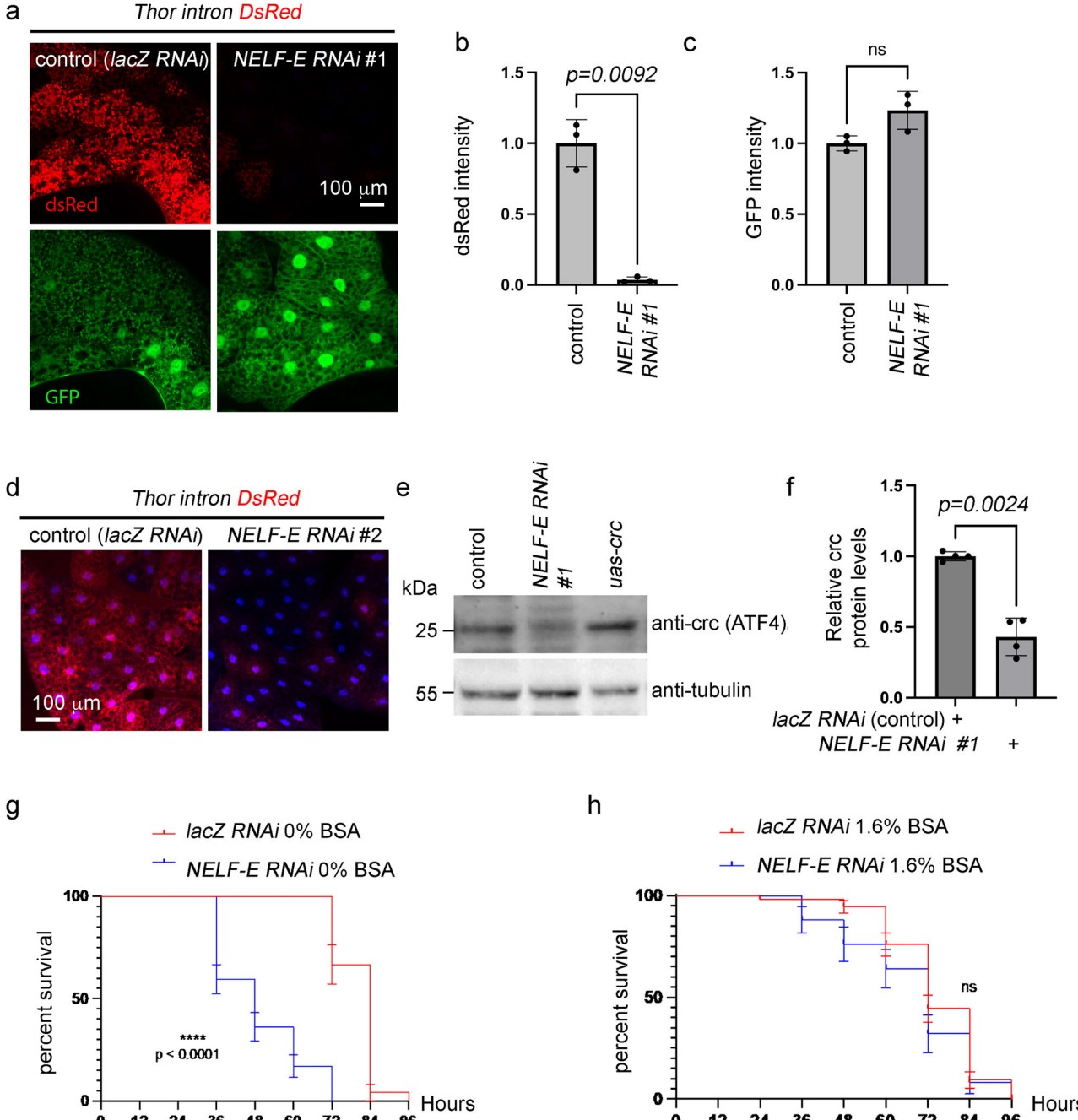

**Fig. 5 | _NELF-E_ RNAi reduces crc (ATF4) protein levels and renders flies vulnerable to protein restriction in the diet.** The indicated RNAi lines were driven to fat body cells using the _dcg-Gal4_ driver. **a** _Thor_ ᶦⁿᵗʳᵒⁿ-_DsRed_ expression (red) in the third instar larval fat body after the knockdown of _lacZ_ (control) or _NELF-E_ (VDRC # 21009 line, here indicated as #1) is shown. The control _dcg-Gal4 > UAS-GFP_ signal (green) are shown in the lower panel. Quantification of the DsRed signal (**b**) and the GFP signal (**c**) across three biological replicates ($n = 3$) shows that _NELF-E_ RNAi's effect is specific to the crc target reporter _Thor_ ᶦⁿᵗʳᵒⁿ-_DsRed_. Two-tailed Welch's _t_ test was used to assess statistical significance. Data presented are mean values +/− SD. **d** An independent RNAi line targeting _NELF-E_ (BDSC #32835 labeled as #2) also reduces _Thor_ ᶦⁿᵗʳᵒⁿ-_DsRed_ expression. Similar fluorescence patterns were consistently observed across multiple independent samples (_lacZ_ RNAi $n = 3$, _NELF-E_ RNAi (BDSC 32835) $n = 4$). **e** Anti-crc (ATF4) western blot from control, _NELF-E_ RNAi (VDRC #21009), and _crc_ overexpressing fat body extracts (top gel). Anti-tubulin

blots are shown as loading controls (bottom gel). **f** Quantification of relative crc band intensities in samples expressing _lacZ_ RNAi ($n = 4$ biological replicates) or _NELF-E_ RNAi ($n = 4$ biological replicates). Two-tailed Welch's _t_ test was used to assess statistical significance. Data presented are mean values +/− SD.
**g**, **h** Kaplan–Meier survival curves showing estimated survival probability ± standard error (SE) of one-day-old adult flies under the indicated conditions. **g** _NELF-E_ RNAi flies (VDRC #21009) ($n = 47$) showed a significant reduction in survival as compared to controls (_lacZ_ RNAi, which is also shown in Fig. 1) when reared without any nutrients. **h** When reared with a protein-only diet (1.6% BSA), no significant difference was seen between _NELF-E_ RNAi ($n = 25$) and control flies. Log-rank analysis (two-sided) was used to assess statistical significance. Log-rank (two-sided) tests were used to assess the statistical significance in (**i**, **j**). _p_- values are listed, and ns indicates not significant.

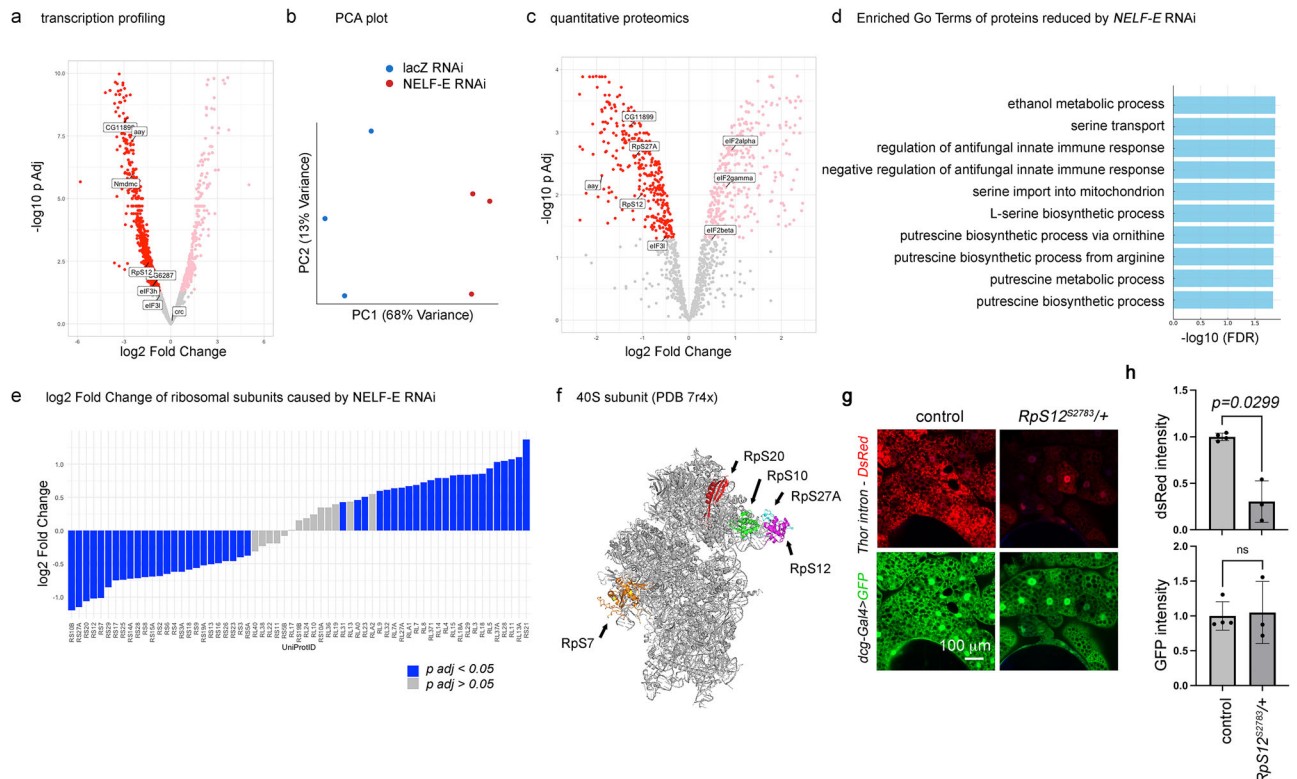

**Fig. 6 | *NELF-E* RNAi changes the levels of metabolic enzymes and ribosome subunits. a** A volcano plot of gene expression changes caused by *NELF-E* RNAi in larval fat body samples, with *lacZ* RNAi serving as the control, across three biological replicates per genotype (*n* = 3). Labeled are enzymes that mediate serine biosynthesis (*CG11899, aay, CG6287*), one-carbon metabolism (*Nmdmc*), and translation mediators (*RpS12, eIF3h*). Differential gene expression was computed via the Seq-N-Slide pipeline (sns) using DESeq2 with its default two-sided Wald test on a negative-binomial generalized linear model; *p*-values were adjusted with the Benjamini–Hochberg method (as described in Fig. 2a). **b** A PCA plot of three independent samples from *lacZ* (control) and *NELF-E* RNAi. **c** A volcano plot of proteomic changes caused by *NELF-E* RNAi in the fat body of three biological replicates per genotype (*n* = 3). Labeled are enzymes mediating serine biosynthesis (CG11899, aay) and those that mediate mRNA translation (RpS12, RpS27A, eIF3l, eIF2alpha, eIF2beta, eIF2gamma). Statistical significance was determined using a two-sided moderated *t* test (limma) with FDR correction (fdrtool) as implemented

in the DEP2 R package (**d**) Enriched GO terms of the proteins reduced by *NELF-E* RNAi. **e** Changes in the levels of Ribosome subunits by *NELF-E* RNAi in the experiment described in (**c**). Those proteins with significant changes (*p adjusted < 0.05*) are labeled in blue. Significantly reduced ribosomal proteins are all part of the 40S subunit (RpS), while many 60S subunit proteins (RpLs) were detected at higher levels. Downregulated ribosome proteins include RpS10b (*p* = 0.0024), RpS12 (*p* = 0.0097), RpS 20 (*p* = 0.0027), and RpS27A (*p* = 0.0024). Significance testing was performed as described in (**c**). **f** Modeling the position of most strongly reduced RpS subunits based on the published human ribosome structure (PDB 7r4x). Three subunits are adjacent to each other. **g** *Thor*^intron^-*DsRed* (red) and *dcg-Gal4 > UAS-GFP* expression (green) in control and *RpS*^S2783^/+ larval fat body. **h** Quantification of DsRed and GFP average pixel intensities of biological replicates prepared from *w*^1118^ control samples (*n* = 4) or *RpS*^S2783^/+ samples (*n* = 3) in (**g**). Data presented are mean values +/− SD. Two-tailed Welch's *t* test was used to assess statistical significance. *p*-values are indicated. ns = not significant.

3-Levels Brightness, 8000 LUX Tracing Light Pad for Children, VKTEKLAB) on the boxes and adjusted the intensity to be 211-259 lux in the no-lid box. The deep pseudopupil (Dpp) in living flies was observed under SMZ1500 (Nikon) under blue light. The observation was done from day 10 to day 28 AE. The survival data was analyzed using "GraphPad Prism 10" (GraphPad Software).

## Confocal microscopy and Immunohistochemistry

Freshly dissected larval tissue was fixed by incubating in 4% PFA for 20 min, and then washed in 0.1% PBS-T three times, 5–10 min each wash. For images displaying samples expressing *Thor*^intron^-*DsRed* and *UAS-GFP*, tissues were then suspended onto whole mount slides in either a solution of 50% glycerol containing DAPI or Vectashield anti-fade mounting medium for fluorescence with DAPI (H-1200). For images displaying samples expressing *crc5'-dsRed*, tissues were labeled with anti-DsRed (Takara 632496, 1:1000) and anti-Rh1 (DSHB 4C5 antibody 1:200) for 1 h. Samples were then washed three additional times before incubated with 546 nm secondary antibodies (1:500) for one hour. After final washes, the tissues were mounted for imaging. Confocal imaging was performed with LSM 700 using a 20x objective lens. Signal intensity was quantified in ImageJ by measuring red

(*Thor*^intron^-*DsRed*, *crc5'-DsRed*) or green (*UAS-GFP*) fluorescence over the range defined by DAPI-stained nuclei.

## RT-qPCR

RNA from the *Drosophila* larval fat body was isolated with TRIzol (Thermo Fisher Scientific) following the manufacturer's instructions. Unless otherwise stated, each biological replicate for RT-qPCR and RNA-seq experiments was composed of RNA derived from 10 male *Drosophila* larval fat bodies for all genotypes. For sample homogenization, pestles for 1.5 mL microcentrifuge tubes (USA Scientific, 1415–5390) were used. Maxima H Minus Reverse Transcriptase was used to generate cDNA from 500 ng RNA, which was subsequently used to perform qPCR with GreenPower SYBR® Green PCR Master Mix (Invitrogen). Relative mRNA levels were determined by using the delta delta CT method to measure cycle threshold, normalized to products generated using primers for *Rp49* or *Rpl15*. The primers for *Thor* and *ATF4* are previously recorded elsewhere. Other primers as follows: Rp49-F 5'- AGATCGTGAAGAAGCGCACCAAG-3'; Rp49-R 5'- CACCAGG AACTTCTTGAATCCGG- 3'; Rpl15-F 5'-AGGATGCACTTATGGCAAGC-3'; Rpl15-R 5'- GCGCAATCCAATACGAGTTC-3'; 4EHP-F 5'- CAGCGA TGTGGATAATCAG-3'; 4EHP-R 5'- GAGAACCAGAGGCAGTAT-3'; NELFE-

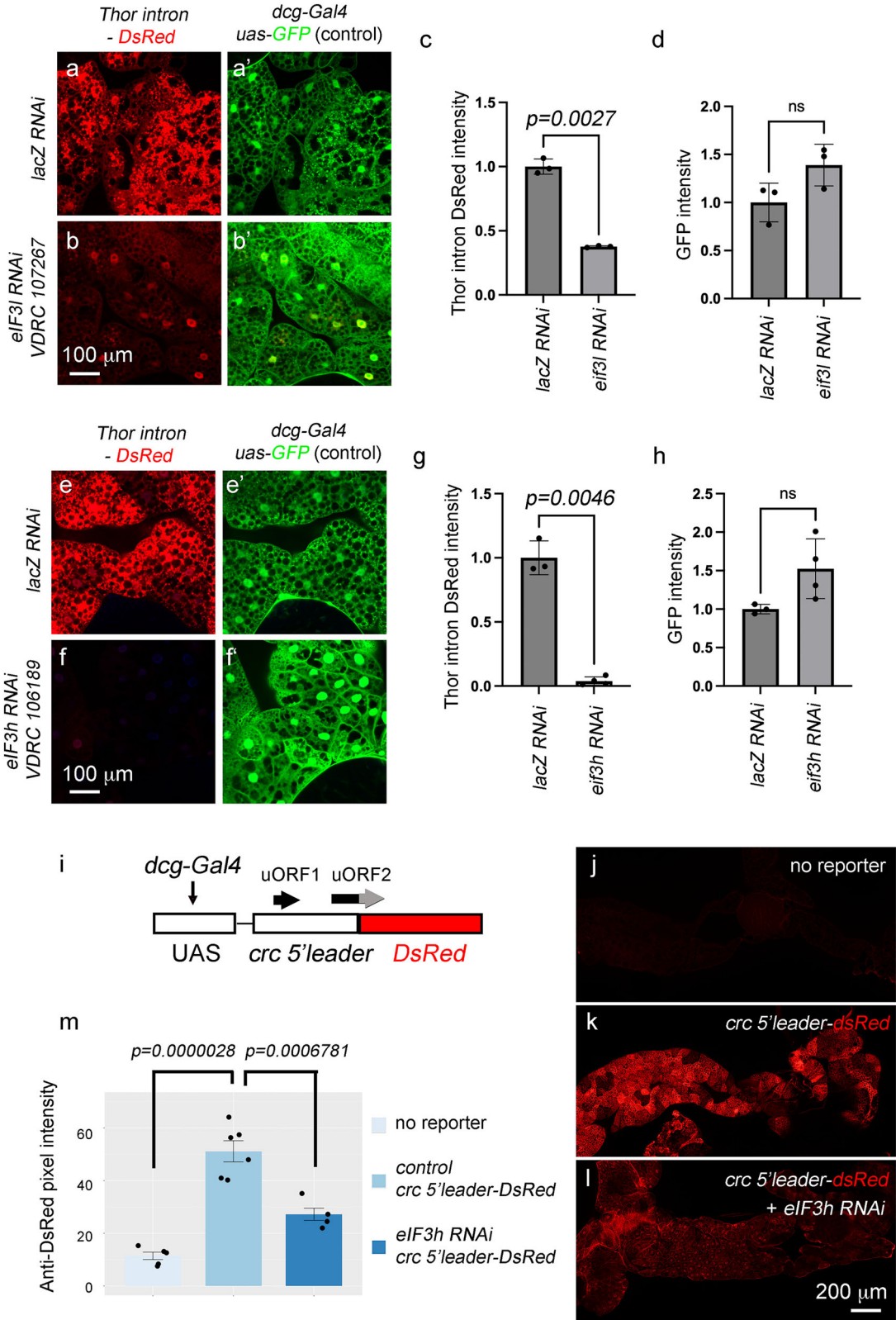

F 5'- TTCATGGCAAGAATGTGAATGG-3'; NELFE-R 5'- GATTTGCTGGC GGCAATAG-3'. eIF3h-F 5'- CTTCAAGCAGGATACGGAGAAG-3'; eIF3h-R 5'-GTTTGATCACTTCCTCCTCTGG-3'.

## RNA-seq and data analysis

RNA from the *Drosophila* larval fat body was isolated with TRIzol following the manufacturer's instructions. The extracted RNA was then treated with DNase (Turbo DNA free kit) to remove contaminant genomic DNA, and then ethanol precipitated. The samples were vacuum dried, resuspended in dH$_2$O and sent to the NYU Genome Technology Center (RRID: SCR_017929) for RNA sequencing using the Illumina NovaSeq 6000 platform using Sp100 Flow Cell v1.5. Triplicate samples for each genotype were analyzed. The cDNA library was prepared from polyA-containing mRNA. We followed Igor Dolgalev's Seq-

**Fig. 7 | *eIF3l* and *eIF3h* are required for physiological crc (ATF4) signaling.**
**a–h** *Thor* $^{intron}$-*DsRed* signal (red in **a**, **b**, **e**, **f**) and the control *dcg-Gal4, UAS-GFP* fluorescence (green in **a'**, **b'**, **e'**, **f'**) in larval fat bodies with the expression of the indicated RNAi lines. *lacZ* RNAi (**a**, **e**) was used as controls. VDRC 107267 was used to knock down *eIF3l,* and VDRC 106189 was used to target *eIF3h*.
**c**, **d**, **g**, **h** Quantification of DsRed (**c**, **g**) and GFP pixel intensities (**d**, **h**) from three biological replicates (*n* = 3) for each of the indicated genotypes. Data presented are mean values +/− SD. Two-tailed Welch's *t* test was used to assess statistical significance. *p*-values are indicated. ns = not significant. **i** A schematic diagram of the *crc 5'UTR-dsRed* reporter. The transgene is expressed through the *dcgGal4/UAS* system. Black arrows above the transgene indicate uORFs. uORF2 overlaps with the

DsRed ORF (red), but in a different reading frame. The gray part of the uORF2 symbolizes changes in the uORF2 coding sequence. **j–l** Anti-DsRed immunolabeling (red) of late larval fat body does not detect signals in a control fat body without the reporter (**j**), but shows reporter activity in response to *crc-5'leader-DsRed* expression (*dcg-Gal4, UAS-crc-5'leader-DsRed*)(**k**). This reporter signal is suppressed when *eIF3h* RNAi (VDRC 106189) is co-expressed (**l**). **m** Quantification of the DsRed pixel intensities of control reporter (no RNAi) samples (*n* = 6 biological replicates), reporter + *eIF3h* RNAi samples (*n* = 4 biological replicates) and no reporter (*n* = 5 biological replicates). Data presented are mean values +/− SE. Ordinary one-way ANOVA and post-hoc Tukey's HSD were used for statistical analysis. ns indicates not significant.

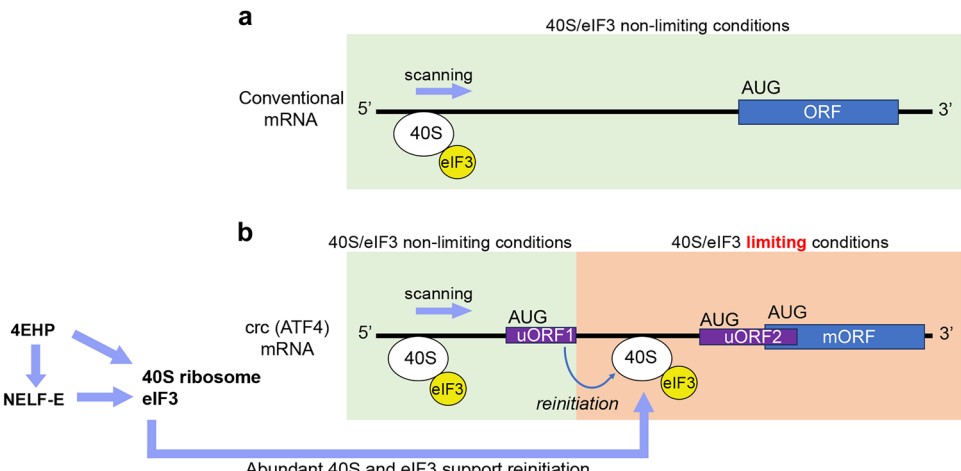

**Fig. 8 | A model of crc (ATF4) regulation by 4EHP and NELF-E. a** Translation of a conventional mRNA with a single Open Reading Frame. The 40S ribosome is depicted as a white oval, and the eIF3 complex in yellow. Other translation initiation factors are not shown. The 40S/eIF3 complex forms at the 5' end and scans the mRNA in search of an AUG start codon. Neither 40S nor eIF3 is limiting under these conditions. **b** Translation of *crc* (*ATF4*) mRNA with two upstream ORFs (uORF1 and uORF2) that precede the main ORF (mORF). Like other mRNAs, the 40S/eIF3

complex scans the mRNA from the 5' end. Translation elongation and termination of uORF1 is predicted to cause 40S/eIF3 dissociation from the mRNA. The limiting amounts of 40S/eIF3 complex that remain associated after uORF1 translation mediates reinitiation of translation at downstream ORFs. Because *4EHP* and *NELF-E* are required to produce abundant levels of 40S and eIF3 subunits, loss of *4EHP* or *NELF-E* impairs the translation of the crc (ATF4) mORF significantly.

N-Slide automated workflow (https://github.com/igordot/sns) to process the fastq files. The DESeq2 package (R v3.6.1) was used for differential gene expression analysis between groups of samples. We filtered the data to analyze transcripts with baseMean values above 350. Normalized cpm counts were used to generate heatmaps.

### Identification of 4EHP-binding RNAs through TRIBE
Targets of RNA-binding proteins discovered by editing (TRIBE) experiments were performed following published protocols[75]. Specifically, we generated an in-frame fusion between 4EHP's coding sequence and the catalytic domain of ADAR through gene synthesis. As a negative control, we generated an equivalent construct with the W114A mutation of 4EHP that disrupts its mRNA cap-binding domain. These sequences were subcloned into pUAST-attB, which were injected to generate targeted insertion lines on the 2$^{nd}$ chromosome. The transgenes were expressed in the larval fat body using the *dcg-Gal4* driver, and the RNAs were sequenced through the NYU Genome Technology Core (see above) to generate fastq files. Preprocessing/adapter trimming was performed using FLEXBAR (https://github.com/seqan/flexbar). Read mapping was done using HiSAT2 version 2.1.0.

JACUSA was used to identify genomic positions where base frequency distributions differ substantially in RNA-DNA (RDD) or RNA-RNA (RRD) comparisons. For the identification of ADAR-edited sites, we followed the Dieterich lab TRIBE workflow (https://github.com/dieterich-lab/tribe-workflow). We specifically considered as 4EHP

targets those that showed preferential A to G conversion by wild-type 4EHP-ADAR as compared to the negative control (4EHP $^{W114A}$ mutant-ADAR). The binding score was defined as a log-likelihood ratio of A to G change on specific nucleotides between the two conditions.

### Antibodies and western blots
Protein extracts were prepared from larval fat bodies with either RIPA buffer or the TRIzol reagent according to the manufacturer's instructions. Protein was resuspended in 1% SDS and quantitated using Pierce BCA Protein Assay. SDS-PAGE was run on approximately 3-microgram samples and transferred onto PVDF (Immobilon-P, IPVH00010) or nitrocellulose (Bio-rad, 1620115) membranes. Membranes were blocked using 5% milk in TBS-T or using SuperBlock blocking buffer in TBS (thermo scientific, 37535). Primary antibodies used include guinea pig anti-crc (ATF4)[34], anti-phospho-eIF2α (1:500, Cell Signaling 9721S), anti-beta tubulin (Covance #MMS-410P), anti-NELF-E[70], anti-DsRed (Takara), anti-Rh1 (DSHB). Secondary antibodies containing HRP (1:5000-1:10000 dilution) were used to image bands, with captured via ChemiDoc or developed using autoradiography sheets. Band intensity was then measured using ImageJ.

### Quantitative proteomics
For each genotype, triplicate samples were analyzed. Protein extracts were prepared from ten male larval fat bodies with the Total Protein Extraction Kit for Adipose Tissue/Cultured Adipocytes (AT-022, Invent Biotechnologies Inc) following the manufacturer's instructions. 10

microliters from each sample was diluted two fold with a buffer consisting of 5% SDS, 20 mM CAA, 20 mM TCEP, 100 mM Tris (pH = 8) and incubated 15 minutes at 90 °C. 10 microliters of magnetic SP3 beads suspension (5% solids) was added, and the proteins were precipitated on beads by two fold dilution with ethanol. The beads were then washed three times with 150 microliter of 85% ethanol, followed by digestion in 100 microliter of buffer containing 0.2 micrograms of trypsin for 8 h at 37 °C (1000 rpm). The resulting peptides were loaded on Evosep Pure C18 tips and analyzed in DIA mode on QExactive HF-X coupled to Evosep One LC system (88 min LC gradient). The resulting MS RAW data were analyzed in Spectronaut using directDIA search mode. Quantification was done on the MS2 level.

### Reporting summary

Further information on research design is available in the Nature Portfolio Reporting Summary linked to this article.

## Data availability

All RNA-seq and TRIBE sequence files are available through the NIH GEO (accession number GSE309407). The raw proteomics data is available through the MassIVE repository (dataset ID MSV000099506). Source data are provided in this paper.

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

## Acknowledgements

We thank Erika Bach, Jessica Treisman, Jean-Yves Roignant, David Levy, Ian Mohr, and Robert. Schneider for helpful comments on the project, and Park Cho-Park, David Gilmour, and ChinTong Ong for sharing reagents. We especially thank Min-Ji Kang, who provided the newly generated guinea pig anti-crc. This project was supported by NIH grants R01EY020866, R35GM148357, R01NS120488 (to H.D.R.), and T32GM136542 (to K.W.). We thank the NYU Langone's Metabolomics Laboratory, Proteomics Laboratory, and the Genome Technology Center for performing the metabolic and gene expression profiling experiments, which were partially supported by the Cancer Center Support Grant P30CA016087.

## Author contributions

K.W. and H.D.R. conceived the project, analyzed the data, and wrote the manuscript with the inputs from the other authors. K.W. performed most of the experiments of this study. Additional contributions include H.K.'s quantitative proteomic analysis and retinal degeneration experiments, H.J. and D.V.'s participation in the RNAi screen together with K.W., and H.D.R.'s experiments related to *parkin* and *crc5'-DsRed*. C.D. bioinformatically analyzed the TRIBE results. H.D.R. supervised the overall project.

## Competing interests

The authors declare no competing interests.
