## [Transparent Peer Review file · Nature Communications]

4EHP and NELF-E regulate physiological ATF4 induction and proteostasis in disease models of *Drosophila*

Corresponding Author: Professor Hyung Don Ryoo

Version 0:

Reviewer comments:

Reviewer #1

(Remarks to the Author)

In this study, the authors discovered novel factors involved in the regulation of ATF4, by genetic screening and TRIBE analysis. It is well established that phosphorylation of eIF2 α is critical for ATF4 activation but other regulators for ATF4 induction has been understudied. Understanding these mechanisms could significantly contribute to elucidating the causes of diseases related to ATF4, making the findings in this paper particularly valuable.

While the necessity of 4EHP, NELF-E, and eIF3h for ATF4 induction has been well demonstrated, there are some data and further analyses needed to clarify the authors' claims.

Major Points:

1 Figures 1, 5, and 7: The manuscript claims that 4EHP, NELF-E, and eIF3h are necessary for ATF4 induction, but experimental data is required to rule out the potential impact on overall development and translation as suppression of translation could decrease ATF4 and the dsRed reporter. The authors need to quantify how loss of function of these genes impact developmental timing (e.g., pupation timing) and general translation capacity. It is also interesting to quantify the intensity of UAS-driven GFP to demonstrate that the suppression of the dsRed reporter is not merely due to reduced translation (similar experiments conducted for eIF3h in Fig. 7b-d should be performed for 4EHP and NELF-E).

2 Figures 1 and 4: As far as it is conducted, I could not find the data to negate the possibility that 4EHP regulates NELF-E at the transcriptional level. If NELF-E transcript levels do not change, it would suggest translational control, but if they do decrease, it may indicate involvement in RNA stability.

3 Figures 1, 4, and 6: The authors need to clarify whether 4EHP, NELF-E, and eIF3h exhibit any changes in expression or activity when ATF4 is physiologically activated (in the late third instar larva). Are they consistently expressed, and do they work in combination when eIF2 α phosphorylation occurs? Do these proteins have any instructive function for ATF4 activation?

5 Figure 2: It is essential to demonstrate whether the metabolic control by 4EHP and NELF-E is through ATF4. Please analyse the data from P7, 4EHP[C53] mutant (or *dgc*>4EHP RNAi) with those from *dgc*>*crc* RNAi. Also, simultaneous manipulation of 4EHP and ATF4 may better provide evidence to show that they are at the same pathway.

6 Figure 2: Changes in one carbon metabolism is interesting but not sure what it means. How much the changes in the metabolism driven by ATF4 contribute to alterations in physiology such as starvation resistance? In this context, the authors may also perform experiments to manipulate amino acid levels in the diet or in the body.

Minor Points:

1 Figure 1a: It would be helpful to mention how the 185 genes selected for screening were chosen.

2 Figures 1d-g: Including red text for dsRed and blue text for DAPI in the figures would enhance clarity.

3 Figures 1c-k: Clarify whether the genetic background of the 4EHP[C53] mutant or KD lines is the same as that of controls, as metabolic levels, transcriptomes, starvation resistance, and lifespan can be significantly influenced by genetic background. If backcrossing was done, this should be stated in the methods; if not, this should be considered in the discussion. The same applies to NELF-E and eIF3h KDs.

4 Figure 3c: In the Park mutant model, does KD of ATF4 restore survivability similarly to the 4EHP mutant?

5 Figure 7: A summary figure of the key findings would be beneficial.

6 Discussion, P17, L354-359: Discussion of lethality regarding KD of eIF3 subunits other than eIF3h should be included in the results section, with data provided in the supplementary materials.

7 P19: It would be useful to clarify how the VDRC and BDSC lines of UAS-4EHP RNAi are differentiated in the text.

Additionally, the BDSC #32835 reference for NELF-RNAi used in Fig. 5 appears to be missing.

Reviewer #2

(Remarks to the Author)

Reviewer #3

(Remarks to the Author)

Walsh et al. explored previously unknown pathways of ATF4 translational control in *Drosophila*. They report an ATF4 regulatory network consisting of eIF4E-Homologous Protein (4EHP), NELF-E, and eIF3. In brief, expression of 4EHP, representing a mRNA cap-binding protein, is required for translation of NELF-E mRNA, a negative regulator of pol II-mediated transcription, and these two proteins combined ensure normal ATF4 signaling in the *Drosophila* larval fat body enabling for example normal production of amino acid metabolites of the Serine-One-Carbon pathway. As a proof of principle, loss of 4EHP, preventing ATF4 induction during stress, suppressed the *Drosophila* Parkin (Park) mutants that serve as a model for a rare familiar form of Parkinson's Disease, and that are characterized by high levels of ATF4 expression due to activated PERK. The prospective role of eIF3 in this novel pathway was not clear to me.

Overall, this work investigates an interesting phenomenon, however, I think it is only half done. In particular, it falls short on the overall proteome analysis (in addition to RNA-seq), does not distinguish whether the observed changes in mRNA levels are due to in/decreased mRNA expression or in/decreased mRNA instability, lacks several important controls, and arrives to several conclusions that, in my humble opinion, are not supported by the data. A detailed, hopefully constructive critique follows.

Major:

1. The introduction, and especially the last paragraph, is very confusing as written. It should be re-written in more logical, simplified way (a schematic of what protein is supposed to do what under what conditions with respect to ATF4 would help a lot too). Also, page 4, line 68; please note that the role of eIF3h in ATF4 translational upregulation in human cell lines was demonstrated in 2017 here (PMID: 28745933).

2. Westerns are in general of poor quality.

- Fig. 1j-upper and 5C; anti-crc signal is not visible for the control samples, darker exposures are required, a loading control is in the case of 5C also very problematic

- Fig. 1j-bottom, 5e-f, and 7h-i; a wrong loading control was picked, the proper one should be anti-eIF2-alpha (or beta or gamma); also, the data are very bumpy and thus inconclusive (it seems that a lot more than 3 replicates will be need to be tested)

3. The reported changes in mRNA expression profiles (e.g. Fig. 6) should be corroborated by comprehensive proteomics to find out if these changes translate into changes in the overall proteome; this is, in my opinion, a critical experiment that can validate all observations and take this report to a whole new level.

4. The last chapter of results on NELF-E and eIF3 seems to me like "sewn with a hot needle". Beginning with figures not referred to in the text, and ending with conclusions that do not seem to be supported by the presented data. I am not saying that eIF3h is not important for ATF4 upregulation, it very likely is, also based on the aforementioned study from human cells, but a lot more should be done to demonstrate it convincingly. Proteomics should be done (see my point 3), all 12 eIF3 subunits should be checked and the resulting data presented. The loss of eIF3h leads to a concomitant loss of 3k and 3l in humans and *Neurospora crassa* (PMID: 27924037, PMID: 27210288). Therefore, eIF3K or L RNAi could be used as an appropriate specificity control. Does eIF3 expression directly depend on the 4EHP/NELF-E axis? To be able to link these three players into a regulatory network, as you did in the abstract, we should simply know more about their causal relationships.

5. Fig. 7e-g. I'm not clear on the design of this "ATF4" reporter construct. uORF2 overlaps with ATF4, how do you get it to overlap with dsRed without disrupting the sequence? Its sequence (including the overlap) seems to play an important role in the overall control of ATF4 as mentioned above (PMID: 28745933). What is the source of the anti-dsRed double band in 7f?

Minor:

The introduction does not cite all relevant data.

- page 4, line 57; ATF4 translational regulation is not only about uORFs (see for example PMID: 38507410), which will become relevant further below.

- page 4, line 62, you can check these studies (PMID: 32589966, PMID: 32589964 PMID: 28119417 PMID: 34352092) that are, in my opinion, equally relevant to this point as those cited.

Fig. 1B. The screen set-up is unclear as depicted.

Page 13, line 274; what is the P-element?

I thank for the opportunity to review this study.
Leos Shivaya Valasek

Reviewer #4

(Remarks to the Author)

In this manuscript, the authors describe novel factors necessary for the induction of Integrated Stress Response (ISR) signaling through ATF4 (*crc* in *Drosophila*). The phosphorylation of eIF2 α in response to stress is well-established for the induction of ISR signaling, leading to expression of *crc*. The expression of *crc* relies on translation re-initiation on the *crc* mRNA after translation of an upstream ORF. Though the eIF3 complex has been implicated previously in this re-initiation process, it is still not well understood how *crc* is expressed as part of ISR signaling. The authors find that 4EHP and NELF-E are necessary for the induction of *crc*, and that the effect of NELF-E is partially mediated through eIF3h, consistent with a role for the eIF3 complex in translation re-initiation. They also present rigorous genetic, metabolic, and transcriptomic data that provides a resource for better understanding the breadth of ISR signaling beyond the focus of this manuscript. These findings are significant because they identify pathways required for *crc* expression in ISR signaling that are independent of eIF2 α phosphorylation.

Despite the interest of these findings, we do have concerns with some of the data interpretation in this study – particularly the implication that the induction of *crc* expression via eIF3h is mediated through both 4EHP and NELF-E, which is not supported by the data presented. In fact, the reduction of many eIF3 subunits is only observed upon knockdown of NELF-E, but not 4EHP, as seen in Table S4. The data is currently presented in a manner that could easily mislead readers to think that all three components are working sequentially, instead of more likely 4EHP and NELF-E being necessary for *crc* expression through parallel mechanisms. In our opinion, this represents mostly a need for some rewriting and clarification/reconsideration of the findings – rather than a problem with the data presented. For example, the first paragraph of the discussion should be revised to make clear that 4EHP and NELF-E are likely involved in partially overlapping but parallel pathways to induce *crc*, with only NELF-E acting through eIF3h. In addition, the inclusion of a model would help make the overall mechanism clearer. We note that there are many genes that are impacted similarly by both NELF-E and 4EHP RNAi, not including eIF3h, that warrant further investigation and discussion – and expanding analysis of these overlapping targets could provide some helpful insight into a potential shared mechanism. Altogether, we think the findings presented in this manuscript are impactful and important, if the conclusions are adjusted to more accurately reflect the data shown.

We have the following specific comments, which are separated into major and minor points:

Major concerns

1. The heatmaps presented in Fig 6c and 7a without any statistics described are misleading because it is unclear whether the differential expression is significant. If the authors chose to remove the 4EHP RNAi from Fig 7a, it should be made clear in the main text that eIF3 subunit transcripts are not significantly downregulated in the 4EHP RNAi. Specifically, the statement on lines 263-264 “These transcripts were also reduced in 4EHP RNAi samples, although to a lesser extent (Table S4)” is incorrect, as they are not significantly reduced based on the adjusted p values shown.
2. A PCA of the RNA-seq samples should be included, especially to compare the two controls and replicate samples. As shown in the heatmaps, it seems that replicate control samples are dissimilar, and it is slightly difficult to tell which control was used in each experiment.
3. Since eIF3h is not reduced by both 4EHP and NELF-E RNAi, a more expansive description (e.g., GO terms) of the overlapping transcripts either reduced or induced (Fig 6a) would be informative. This would be a more balanced approach to describing these data rather than the heatmaps shown of selected genes.
4. For the degenerative disease models (Fig. 3), it is unclear whether these phenotypes are mediated through *crc*. Additional experiments or comparison to previous data may bolster this argument. For instance, a 4EHP and *crc* double mutant in Fig 3d should show a non-compounding effect on ommatidial integrity if they are working in the same cascade. Where possible, comparisons to phenotypes with *crc* knockdown or overexpression would make it easier to conclude that 4EHP and NELF-E loss phenotypes are due to downstream impacts on *crc*. Some of these findings have been published elsewhere, but it is difficult to compare directly without inclusion of these data in the same analysis.
5. The mRNAs identified in the TRIBE experiment should be described more comprehensively. A GO term analysis may be illuminating since there are many interesting metabolic genes identified (Table S2). Since CG18132 may be involved in protein folding in the ER and is the top TRIBE hit, it should be more thoroughly described if mentioned at all.
6. The Western blots were not very clear in places, and the loading seems uneven with poor intensity for many of the target proteins of interest. In addition, it is unclear how the statistical analysis of band intensity comparison were performed. In some cases, all controls appear to have the same value with no error bar (e.g., Fig 4e), which may be a plotting error. The authors may consider a different method of quantifying *crc* protein levels (e.g., epitope tagging, alternative protein extraction strategy without using TRIZOL) if the antibody does not perform well, as in Fig 5c. Since many conclusions hinge on 4EHP and NELF-E impacting *crc* protein levels, high quality Western blots are necessary.
7. The findings from the RNAi screen should be expanded upon, especially given that eIF4EHP is not the top listed suppressor of Thor intron expression in Table S1. A brief description of the other screen hits and a stronger rationale for following 4EHP would be beneficial. Quantitative data should be presented in Table S1 if possible, so the extent to which Thor intron expression is impacted can be assessed. This table (and all others where applicable) should be fully annotated with gene names and FBgn IDs to make these data fully accessible to other researchers.

Minor concerns

1. Sequencing data (RNA-seq and TRIBE) should be deposited to an accessible repository like GEO.
2. A supplemental table describing the metabolic profiling data should be provided.
3. Park13 instead of Park25 seems to be referred to in error or the usage is unclear. For instance, "We found that Park13/ParkD21 adult flies showed intense Thorintron-dsRed reporter expression indicative of strong ATF4 signaling (Figure 3a, b)" is written on line 164, but Park13/ParkD21 flies are not shown in the indicated figure.
4. Supp Fig S6 needs a visible control.
5. In some cases (e.g., Fig 2a) axis text on plots is not readable.

Reviewer #5

(Remarks to the Author)

Version 1:

Reviewer comments:

Reviewer #1

(Remarks to the Author)

I thank the authors to have addressed the concerns and I believe the manuscript is suitable for publication.

Reviewer #2

(Remarks to the Author)

Reviewer #3

(Remarks to the Author)

The authors did a great job and the manuscript improved considerably. Thank you for that. However, I still have a few issues that, IMHO, must be resolved before the manuscript can be considered suitable for publication in this journal.

1) Figs 1j, 5g, 7n, etc. As I noted in my original comment, to use tubulin as a normalization control for the anti-P-eIF2 signal is technically very wrong and any conclusion driven from these experiments concerning the eIF2-independence of the observed effects is dangerously misleading (e.g. lines 347-348; but mainly the closing paragraph 433 – 439!). Moreover, taking into account that the newly added proteomics revealed the RNAi analysis of 4EHP and NELF-E increased levels of all three eIF2 subunits! To examine if the P-status of eIF2-alpha has changed or not under given stress/condition, the anti-P-eIF2 signal has to be normalized only to anti-eIF2alpha (in the worst case to other eIF2 subunits, although they do not always occur in the equimolar ratio). Please check the old scholarly papers by Hinnebusch, Wek, Ron, Sonenberg... It really is a must.

2) I understand that the human SL3 might not be present everywhere. We have seen it ourselves. What my comment was supposed to imply is the following (and I really do not care if the authors cite our most recent paper describing SL3 or not – it is not about this!). Ignoring this uORF2/ATF4 overlap region in the past generated confusing/conflicting results. One way or the other, it is a conserved region, because it is a coding region. Therefore, it is likely, maybe even highly likely, that it contains some mRNA features acting in-trans even in D.m. Have you checked its potential to form stable sec. structures? Your construct in Fig. 7 is responding as expected, but can you guarantee that you are monitoring the physiological response? Perhaps this overlap makes it weaker or stronger? Maybe not, we do not know, but why to repeat the same "mistake"? If anything, the authors should state clearly in the main text that the fact that their construct lacks this region represents a possible limitation to its ability to respond at the full physiological scale. And again, is it perfectly fine to ignore us :)

This thinking brought me back to the title. Considering these and other differences, I guess the title should make a reader aware of the fact that this work explored D.m.; e.g. "4EHP and NELF-E regulate physiological 1 ATF4 induction and proteostasis in disease models of *Drosophila melanogaster*." Or something like this.

3) A few notes to consider:

- there is literature showing that eIF3I and 3k are considered to be inhibitory (not initiation-stimulatory) subunits

- lines 301 – 303; what do you mean by "a compensatory mechanism"? Specific reduction in biogenesis of one ribosomal subunit will naturally lead to an increase of the other subunit.

- 303 – 307; this is really interesting; It seems that the 40S bio defect lies in an impaired formation of the 40S beak. Can you check in Ramakrishan's Science paper (2020) where exactly the positions of 3l and 3k in the 48S PIC were mapped? They can exist as a dimer out of eIF3 and thereby they might be needed for final maturation of the 40S, which would explain a lot. Also, there is an EMBO J. paper from D. Wolf's lab (2022?) on eIF3k and ribosomal proteins with respect to 40S biogenesis that could help too.

- lines 310 – 314; If my assumption is correct, I would be very curious to see if the levels of RPS go down and those of RPL go up in RPS12-S2383/+ in a similar fashion as observed with 4EHP and NELF-E siRNAs. This could nail the mol mech down.

- 318; I would be more accurate: "the post-termination 40S subunits".

- 321; eIF3j is not a bona fide eIF3 subunit, it is only an eIF3-associated factor

- 419; for the sake of collegiality, please also include 26 and 29; this is important to us.

- 425 – 426; for same reason, it would be fair to add that a similar observation was made in humans (31).

Reviewer #4

(Remarks to the Author)

The authors have addressed all of our concerns and the revised manuscript is much improved. The addition of the proteomic data really strengthens this study. Very interesting paper.

Reviewer #5

(Remarks to the Author)

Point-by-point response to reviewer comments

We thank the reviewers for the overall positive review. Below is a summary of our revision in response to specific reviewer comments.

Reviewer #1 (Remarks to the Author)

In this study, the authors discovered novel factors involved in the regulation of ATF4, by genetic screening and TRIBE analysis. It is well established that phosphorylation of eIF2 α is critical for ATF4 activation but other regulators for ATF4 induction has been understudied. Understanding these mechanisms could significantly contribute to elucidating the causes of diseases related to ATF4, making the findings in this paper particularly valuable.

Thank you for the positive assessment.

While the necessity of 4EHP, NELF-E, and eIF3h for ATF4 induction has been well demonstrated, there are some data and further analyses needed to clarify the authors' claims.

Major Points:

1 Figures 1, 5, and 7: The manuscript claims that 4EHP, NELF-E, and eIF3h are necessary for ATF4 induction, but experimental data is required to rule out the potential impact on overall development and translation as suppression of translation could decrease ATF4 and the dsRed reporter. The authors need to quantify how loss of function of these genes impact developmental timing (e.g., pupation timing) and general translation capacity. It is also interesting to quantify the intensity of UAS-driven GFP to demonstrate that the suppression of the dsRed reporter is not merely due to reduced translation (similar experiments conducted for eIF3h in Fig. 7b-d should be performed for 4EHP and NELF-E).

Thank you for the point. To demonstrate that ATF4 reduction is not a consequence of indiscriminate suppression of protein synthesis, we have added control *dcg-Gal4>UAS-GFP* expression data in fat bodies with *4EHP RNAi* (revised Figure 1f'), *NELF-E RNAi* (revised Figure 5a), and *eIF3h RNAi* (revised Figure 7f'). The results clearly show that knockdowns of *4EHP*, *NELF-E*, and *eIF3h* specifically reduces the *crc* (ATF4) reporter but not the control GFP expression. Furthermore, we followed Reviewer #3's suggestion to perform quantitative proteomics in samples with *4EHP RNAi* (Figure 2c) and *NELF-E RNAi* (Figure 6c), and the results do not show a general reduction in the proteome. In fact, many 60S ribosome subunits and translation initiation factor peptides are found at higher levels in the *4EHP* and *NELF-E RNAi* dataset (Figures 2c, S6, 6c, 6e). We now include a Supplemental Figure showing that the knockdown of *4EHP* or *NELF-E* delays *Drosophila* development (Figure S1, S8e), but the results with *dcg-Gal4>UAS-GFP* and quantitative proteomics indicate that the effect of *4EHP* and *NELF-E RNAi* on *crc* is specific and not due to indiscriminate suppression of all protein synthesis.

2 Figures 1 and 4: As far as it is described, I could not find the data to negate the possibility that 4EHP regulates NELF-E at the transcriptional level. If NELF-E transcript levels do not change, it would suggest translational control, but if they do decrease, it may indicate involvement in RNA stability.

Thank you for the point. The revised manuscript now has a new Figure 4e, which shows that *NELF-E* transcript levels do not change significantly in the *4EHP RNAi* condition as revealed in the RNA-seq dataset.

3 Figures 1, 4, and 6: The authors need to clarify whether 4EHP, NELF-E, and eIF3h exhibit any changes in expression or activity when ATF4 is physiologically activated (in the late third instar larva). Are they consistently expressed, and do they work in combination when eIF2 α phosphorylation occurs? Do these proteins have any instructive function for ATF4 activation?

Thank you for the point. We have added a new Figure S16 showing that ATF4 (*crc*) knockdown in the third instar larval fat body does not change the transcript levels of *4EHP*, *NELF-E*, and *eIF3h*, arguing against the idea that physiologically activated ATF4 regulates their expression. To determine whether these factors have instructive function for ATF4 activation, we overexpressed *4EHP* (Figure S3a, b), *NELF-E* (Figure S8a, b), and *eIF3h* (Figure S15a, b), but they were not sufficient to increase ATF4 reporter expression. The results are not surprising as these factors operate within a larger protein complex that requires other subunits for their function.

5 Figure 2: It is essential to demonstrate whether the metabolic control by 4EHP and NELF-E is through ATF4. Please analyse the data from P7, 4EHP[C53] mutant (or *dcg>4EHP* RNAi) with those from *dcg>crc* RNAi. Also, simultaneous manipulation of 4EHP and ATF4 may better provide evidence to show that they are at the same pathway.

Following the suggestion, we performed metabolic profiling in *crc* (ATF4) RNAi fat body, but we did not obtain similar profiles with the 4EHP mutants. We note that metabolic profiles are susceptible to small differences in inter-organ transport or the metabolic flux. Because of the imperfect data, we deleted conclusions attributing the metabolic change to ATF4 signaling (pages 7, 8). The revised manuscript now focuses on similarities in gene expression rather than metabolic profiles. We removed the word “amino acid metabolism” from the title, and the metabolomics data have been moved to a Supplemental Figure (Fig. S4). The metabolic profiling data in Figure 2 were replaced with gene expression profiling and quantitative proteomics data as requested by Reviewer #3. The revised text now centers around describing the 4EHP loss-of-function phenotype with the following logical flow: We comment that the enriched GO terms of proteins reduced by 4EHP RNAi are mostly associated with metabolism (Fig. 2d), that most significant changes in the metabolome are related to the serine-one-carbon pathway (Fig. S4), and consistently, the flies are vulnerable to protein starvation (Figure 2).

6 Figure 2: Changes in one carbon metabolism is interesting but not sure what it means. How much the changes in the metabolism driven by ATF4 contribute to alterations in physiology such as starvation resistance? In this context, the authors may also perform experiments to manipulate amino acid levels in the diet or in the body.

To establish an association with ATF4, we have added new data showing that *crc* (ATF4) RNAi renders flies more vulnerable to starvation, and this vulnerability is suppressed upon providing a protein-only diet (see revised Figure 2h, k).

Minor Points:

1 Figure 1a: It would be helpful to mention how the 185 genes selected for screening were chosen.

The revised text page 5 now states that the 183 RNAi lines targeted either known translation regulators or the *Drosophila* homologs of ribosome-associated proteins (page 5 lines 100 - 101, ref. 38).

2 Figures 1d-g: Including red text for dsRed and blue text for DAPI in the figures would enhance clarity.

Thank you for the point. We now mark dsRed and DAPI with colored fonts in Figures 1c and d.

3 Figures 1c-k: Clarify whether the genetic background of the 4EHP[C53] mutant or KD lines is the same as that of controls, as metabolic levels, transcriptomes, starvation resistance, and lifespan can be significantly influenced by genetic background. If backcrossing was done, this should be stated in the methods; if not, this should be considered in the discussion. The same applies to NELF-E and eIF3h KDs.

Thank you for the point. All flies were in the w1118 background. While they had not been backcrossed to a common strain, 4EHP RNAi results (Fig. 1c-f, 2g, j) were validated with 4EHP^{CP53} (Fig. S2, Fig. 2f, i), NELF-E RNAi results were validated with two independent RNAi lines (Fig. 5a, d), and eIF3h were also validated with two independent RNAi lines (Fig. 7f, S12). Following the reviewer's suggestion, we also included a statement in the Discussion that "the *Drosophila* lines used in this study were not backcrossed to a common parental strain, but key results were all corroborated with independent RNAi lines or with classical mutant alleles (page 17 lines 390 - 392).

4 Figure 3c: In the Park mutant model, does KD of ATF4 restore survivability similarly to the 4EHP mutant?

A strong loss of *crc* (ATF4) can lead to proteostasis defects and lethality. However, we have added Figure S7 showing that the *crc* -/+ background leads to a moderate (and statistically significant) extension of Park mutant lifespan.

5 Figure 7: A summary figure of the key findings would be beneficial.

A summary diagram has been added as the new Figure 8.

6 Discussion, P17, L354-359: Discussion of lethality regarding KD of eIF3 subunits other than eIF3h should be included in the results section, with data provided in the supplementary materials.

Thank you for the point. We've added the lethality data as the new Figure S11.

7 P19: It would be useful to clarify how the VDRC and BDSC lines of UAS-4EHP RNAi are differentiated in the text. Additionally, the BDSC #32835 reference for NELF-RNAi used in Fig. 5 appears to be missing.

We have now added information regarding the 4EHP RNAi lines in the legends of Figures 1, 2, 4, and the reference to NELF-E RNAi #1 (VDRC line) in the revised legends of Figures 4 and 5.

Reviewer #2 (Remarks to the Author):

Reviewer #3 (Remarks to the Author):

Walsh et al. explored previously unknown pathways of ATF4 translational control in *Drosophila*. They report an ATF4 regulatory network consisting of eIF4E-Homologous Protein (4EHP), NELF-E, and eIF3. In brief, expression of 4EHP, representing a mRNA cap-binding protein, is required for translation of NELF-E mRNA, a negative regulator of pol II-mediated transcription, and these two proteins combined ensure normal ATF4 signaling in the *Drosophila* larval fat body enabling for example normal production of amino acid metabolites of the Serine-One-Carbon pathway. As a proof of principle, loss of 4EHP, preventing ATF4 induction during stress, suppressed the *Drosophila* Parkin (Park) mutants that serve as a model for a rare familiar form of Parkinson's Disease, and that are characterized by high levels of ATF4 expression due to activated PERK. The prospective role of eIF3 in this novel pathway was not clear to me.

Overall, this work investigates an interesting phenomenon, however, I think it is only half done. In particular, it falls short on the overall proteome analysis (in addition to RNA-seq), does not distinguish whether the observed changes in mRNA levels are due to in/decreased mRNA expression or in/decreased mRNA instability, lacks several important controls, and arrives to several conclusions that, in my humble opinion, are not supported by the data. A detailed, hopefully constructive critique follows.

Major:

1. The introduction, and especially the last paragraph, is very confusing as written. It should be re-written in more logical, simplified way (a schematic of what protein is supposed to do what under what conditions with respect to ATF4 would help a lot too). Also, page 4, line 68; please note that the role of eIF3h in ATF4 translational upregulation in human cell lines was demonstrated in 2017 here (PMID: 28745933).

Thank you for the point. We re-wrote the last paragraph of the introduction to enhance clarity. We now cite PMID 28745933 (ref. 31) following the reviewer's suggestion.

2. Westerns are in general of poor quality.

- Fig. 1j-upper and 5C; anti-crc signal is not visible for the control samples, darker exposures are required, a loading control is in the case of 5C also very problematic

Following the suggestion, we replaced the blot in Figure 5e with an improved new blot. We also enhanced the contrast of the anti-crc blot in Figure 1. Please note that anti-crc under physiological stress (in a normally developing fat body) is naturally very weak as compared to those in other studies that use extreme stress imposed by tunicamycin or thapsigargin treatment. Our blot aimed to show this by comparing its weak band to that of crc overexpression (Figure 1j lane 3).

- Fig. 1j-bottom, 5e-f, and 7h-i; a wrong loading control was picked, the proper one should be anti-eIF2-alpha (or beta or gamma); also, the data are very bumpy and thus inconclusive (it seems that a lot more than 3 replicates will be need to be tested)

We followed this reviewer's suggestion to perform quantitative proteomics (see points below), and the results indicate that all three eIF2 subunit levels "increase" in response to the knockdown of 4EHP or NELF-E (Figures 2c, 6c, Supplemental Table S3). For example, eIF2alpha nearly doubles (row 341 in the Table S3), eIF2beta is detected at 2.6 fold higher (row 342 in Table S3), and eIF2gamma is 2.4 fold higher than controls (row 517 of Table S3) in 4EHP RNAi fat body – a reason why we couldn't use them as loading controls. We replaced the bumpy blot of Figure 5e with a new blot.

We performed additional replicate western blots, and this is now indicated in the quantified graphs of Figures 1k, l, and 5f. Please note that our ATF4 reporter, Thor intron-dsRed, provides a robust independent corroboration of our conclusions regarding ATF4.

3. The reported changes in mRNA expression profiles (e.g. Fig. 6) should be corroborated by comprehensive proteomics to find out if these changes translate into changes in the overall proteome; this is, in my opinion, a critical experiment that can validate all observations and take this report to a whole new level.

Thank you for the point. Following this suggestion, we performed quantitative proteomics with 4EHP and NELF-E RNAi fat bodies (Table S3, S7; Figures 2c, S6, 6c-e). The results revealed that serine biosynthesis enzymes, RpS subunits, and eIF3l are significantly reduced. We have revamped Figures 2 and 6 to incorporate these results.

4. The last chapter of results on NELF-E and eIF3 seems to me like "sewn with a hot needle". Beginning with figures not referred to in the text, and ending with conclusions that do not seem to be supported by the presented data. I am not saying that eIF3h is not important for ATF4 upregulation, it very likely is, also based on the aforementioned study from human cells, but a lot more should be done to demonstrate it convincingly. Proteomics should be done (see my point 3), all 12 eIF3 subunits should be checked and the resulting data presented. The loss of eIF3h leads to a concomitant loss of 3k and 3l in humans and *Neurospora crassa* (PMID: 27924037, PMID: 27210288). Therefore, eIF3K or L RNAi could be used as an appropriate specificity control. Does eIF3 expression directly depend on the 4EHP/NELF-E axis? To be able to link these three players into a regulatory network, as you did in the abstract, we should simply know more about their causal relationships.

Great suggestion. We performed quantitative proteomics, which yielded important insights into the 4EHP-NELF-E pathway. Unlike the case of humans and *Neurospora crassa*, eIF3k, l, and h were not concomitantly lost. We found that while NELF-E regulates the transcripts of many eIF3 subunits, at the proteome level, only eIF3l was commonly reduced ($p_{adj} < 0.05$) in the *Drosophila* fat body with 4EHP or NELF-E RNAi.

We also noted that many RpS subunits are significantly reduced at the proteome level in both 4EHP and NELF-E RNAi fat body. This is notable because a complex between the 40S ribosome and eIF3 is necessary for re-initiation downstream of uORFs (according to Dr. Valasek's studies). We added new data on RpS in the revised Figure 6e.

5. Fig. 7e-g. I'm not clear on the design of this "ATF4" reporter construct. uORF2 overlaps with ATF4, how

do you get it to overlap with dsRed without disrupting the sequence? Its sequence (including the overlap) seems to play an important role in the overall control of ATF4 as mentioned above (PMID: 28745933). What is the source of the anti-dsRed double band in 7f?

The reviewer is correct in that the uORF2 coding sequence changes in the regions that overlap with dsRed, and to highlight this, we have changed the color of that coding sequence to gray in Figure 7f. However, we show in a new Supplemental Figure (Fig. S14) that this reporter is robustly induced by stress and is completely abolished when the eIF2a kinase (in this case, PERK) is lost.

We were not able to figure out the nature of the double band in 7f. To avoid confusion, we replaced those panels with immunohistochemistry, which show signals that strictly depend on the reporter (see Fig. 7j-l).

Minor:

The introduction does not cite all relevant data.

- page 4, line 57; ATF4 translational regulation is not only about uORFs (see for example PMID: 38507410), which will become relevant further below.

The study by Dr. Valasek (PMID 38507410) has determined human ATF4's regulation by a stem loop sequence referred to as SL3. However, that study had investigated SL3 only in mammals, and its relevance in *Drosophila* had not been examined. Below, we show nucleotide (panels A, B) and amino acid sequence alignments (panel C) to highlight that SL3 is not conserved in *Drosophila* *crc*. Therefore, the issue of SL3 and the relevance of the uORF2 coding sequence is not relevant to our manuscript.

A. Human ATF4 (first 20 a.a. coding)

A stem-loop-forming regulatory sequence (SL3)

B. *Drosophila* *crc* (first 23 a.a. coding)

C.

Multiple Protein Alignment for ATF4 ? DIOPT v8.0 More on DIOPT

- page 4, line 62, you can check these studies (PMID: 32589966, PMID: 32589964 PMID: 28119417 PMID: 34352092) that are, in my opinion, equally relevant to this point as those cited.

Thank you for pointing this out. We now cite all those papers while introducing the process of re-initiation on page 4.

Fig. 1B. The screen set-up is unclear as depicted.

To help readers unfamiliar with gene expression methods in *Drosophila*, we now cite Brand and Perrimon (1993) on page 5 line 99, which describes how the Gal4/UAS system of gene expression works.

Page 13, line 274; what is the P-element?

We now indicate on revised page 16 (line 348) that P-element is a transposable element.

Page 17, lines 354-359. This part should be moved to results, I think.

Following the suggestion, we have moved that part to the results section with a new figure (Figure S11).

I thank for the opportunity to review this study.
Leos Shivaya Valasek

Reviewer #4 (Remarks to the Author):

In this manuscript, the authors describe novel factors necessary for the induction of Integrated Stress Response (ISR) signaling through ATF4 (*crc* in *Drosophila*). The phosphorylation of eIF2 α in response to stress is well-established for the induction of ISR signaling, leading to expression of *crc*. The expression of *crc* relies on translation re-initiation on the *crc* mRNA after translation of an upstream ORF. Though the eIF3 complex has been implicated previously in this re-initiation process, it is still not well understood how *crc* is expressed as part of ISR signaling. The authors find that 4EHP and NELF-E are necessary for the induction of *crc*, and that the effect of NELF-E is partially mediated through eIF3h, consistent with a role for the eIF3 complex in translation re-initiation. They also present rigorous genetic, metabolic, and transcriptomic data that provides a resource for better understanding the breadth of ISR signaling beyond the focus of this manuscript. These findings are significant because they identify pathways required for *crc* expression in ISR signaling that are independent of eIF2 α phosphorylation.

Thank you for the positive assessment.

Despite the interest of these findings, we do have concerns with some of the data interpretation in this study – particularly the implication that the induction of *crc* expression via eIF3h is mediated through both 4EHP and NELF-E, which is not supported by the data presented. In fact, the reduction of many eIF3 subunits is only observed upon knockdown of NELF-E, but not 4EHP, as seen in Table S4. The data is

currently presented in a manner that could easily mislead readers to think that all three components are working sequentially, instead of more likely 4EHP and NELF-E being necessary for *crc* expression through parallel mechanisms. In our opinion, this represents mostly a need for some rewriting and clarification/reconsideration of the findings – rather than a problem with the data presented. For example, the first paragraph of the discussion should be revised to make clear that 4EHP and NELF-E are likely involved in partially overlapping but parallel pathways to induce *crc*, with only NELF-E acting through eIF3h. In addition, the inclusion of a model would help make the overall mechanism clearer. We note that there are many genes that are impacted similarly by both NELF-E and 4EHP RNAi, not including eIF3h, that warrant further investigation and discussion – and expanding analysis of these overlapping targets could provide some helpful insight into a potential shared mechanism. Altogether, we think the findings presented in this manuscript are impactful and important, if the conclusions are adjusted to more accurately reflect the data shown.

We have the following specific comments, which are separated into major and minor points:

Major concerns

1. The heatmaps presented in Fig 6c and 7a without any statistics described are misleading because it is unclear whether the differential expression is significant. If the authors chose to remove the 4EHP RNAi from Fig 7a, it should be made clear in the main text that eIF3 subunit transcripts are not significantly downregulated in the 4EHP RNAi. Specifically, the statement on lines 263-264 “These transcripts were also reduced in 4EHP RNAi samples, although to a lesser extent (Table S4)” is incorrect, as they are not significantly reduced based on the adjusted p values shown.

Thank you for this point. Following the reviewer’s suggestion, we removed the old heatmap from Figures 6c and 7a and deleted the comment regarding the reduced expression of specific eIF3 subunits to a lesser extent. We replaced this part with the proteins that were commonly reduced in 4EHP and NELF-E RNAi samples. Specifically, Reviewer #3 had requested that we perform quantitative proteomics with these samples, and the results indicate that multiple RpS subunits and eIF3l are commonly reduced with statistical significance ($p \text{ adj} < 0.05$). Those new results are highlighted in the revised Figure S10.

2. A PCA of the RNA-seq samples should be included, especially to compare the two controls and replicate samples. As shown in the heatmaps, it seems that replicate control samples are dissimilar, and it is slightly difficult to tell which control was used in each experiment.

Following the reviewer’s suggestion, we have now included PCA plots for the RNA-seq results in Figures 2b and Figure 6b. Please note that we decided to separate the two RNA-seq results because NELF-E RNAi was done at 20 °C, and 4EHP RNAi was performed in 25 °C. The 4EHP RNAi results are now shown in the revised Figure 2, and NELF-E RNAi is shown in the revised Figure 6.

3. Since eIF3h is not reduced by both 4EHP and NELF-E RNAi, a more expansive description (e.g., GO terms) of the overlapping transcripts either reduced or induced (Fig 6a) would be informative. This would be a more balanced approach to describing these data rather than the heatmaps shown of selected genes.

Thank you for the excellent suggestion. Since Reviewer #3 has implied that the proteome change is more relevant than transcript changes, we show enriched GO terms for reduced proteins in Figure 2d (4EHP RNAi samples), Figure 6d (NELF-E RNAi samples), and Figure S10b (proteins commonly reduced in 4EHP and NELF-E RNAi samples). The enriched GO terms in the commonly reduced proteins include “cytoplasmic translation” (Figure S10b), which provides a nice segue into our examination of ribosomal proteins and initiation factors. Thanks to the reviewer’s suggestion, we feel this part of the revised manuscript now has an improved logical flow.

4. For the degenerative disease models (Fig. 3), it is unclear whether these phenotypes are mediated through *crc*. Additional experiments or comparison to previous data may bolster this argument. For instance, a 4EHP and *crc* double mutant in Fig 3d should show a non-compounding effect on ommatidial integrity if they are working in the same cascade. Where possible, comparisons to phenotypes with *crc* knockdown or overexpression would make it easier to conclude that 4EHP and NELF-E loss phenotypes are due to downstream impacts on *crc*. Some of these findings have been published elsewhere, but it is difficult to compare directly without inclusion of these data in the same analysis.

Thank you for the suggestion. We have now included a new Figure S7 to show the effect of *crc* +/- on the Parkin mutants. In addition, we include a new Figure 3d that shows accelerated retinal degeneration of *crc* hypomorphs. We didn’t do the 4EHP and *crc* double mutant analysis because both are hypomorphs (null alleles are lethal), and we expect an additive effect that is difficult to interpret.

5. The mRNAs identified in the TRIBE experiment should be described more comprehensively. A GO term analysis may be illuminating since there are many interesting metabolic genes identified (Table S2). Since CG18132 may be involved in protein folding in the ER and is the top TRIBE hit, it should be more thoroughly described if mentioned at all.

Thank you for the point. Following the suggestion, we added a GO Term enrichment graph in the revised Figure 4c associated with the TRIBE study. We also introduced CG18132 in the revised manuscript as “an uncharacterized protein disulfide isomerase, predicted to assist protein folding within the endoplasmic reticulum (page 12 line 249 -250).”

6. The Western blots were not very clear in places, and the loading seems uneven with poor intensity for many of the target proteins of interest. In addition, it is unclear how the statistical analysis of band intensity comparison were performed. In some cases, all controls appear to have the same value with no error bar (e.g., Fig 4e), which may be a plotting error. The authors may consider a different method of quantifying *crc* protein levels (e.g., epitope tagging, alternative protein extraction strategy without using TRIzol) if the antibody does not perform well, as in Fig 5c. Since many conclusions hinge on 4EHP and NELF-E impacting *crc* protein levels, high quality Western blots are necessary.

Thank you for the point. Following this suggestion, we used the RIPA buffer to extract additional samples, and provide a new Figure 5e (anti-*crc*), and Figures 1j and 5g (phosphor-eIF2alpha). Additionally, we present new graphs showing normalization to the average intensity of control band loading controls, which reflects the variability between controls.

7. The findings from the RNAi screen should be expanded upon, especially given that eIF4EHP is not the top listed suppressor of Thor intron expression in Table S1. A brief description of the other screen hits

and a stronger rationale for following 4EHP would be beneficial. Quantitative data should be presented in Table S1 if possible, so the extent to which Thor intron expression is impacted can be assessed. This table (and all others where applicable) should be fully annotated with gene names and FBgn IDs to make these data fully accessible to other researchers.

Following the suggestion, we annotated Table S1 with gene names and FBgn IDs. The initial hits were not based on quantitative measurements and, therefore, should be considered as preliminary data. We now include that comment in the Table S1 legend. We comment that 4EHP's cap-binding domain may have a "more specific set of target mRNAs." (page 6, line 122). We further add in the revised manuscript that "we decided to further characterize 4EHP due to its possible specificity in gene expression control" (page 6, line 125).

Minor concerns

1. Sequencing data (RNA-seq and TRIBE) should be deposited to an accessible repository like GEO.

Following the suggestion, we have deposited the data in NIH GEO and added the accession number in the Data Availability section.

2. A supplemental table describing the metabolic profiling data should be provided.

Following the suggestion, we have added the raw metabolic profiling data as the new Table S4.

3. Park13 instead of Park25 seems to be referred to in error or the usage is unclear. For instance, "We found that Park13/ParkD21 adult flies showed intense Thorintron-dsRed reporter expression indicative of strong ATF4 signaling (Figure 3a, b)" is written on line 164, but Park13/ParkD21 flies are not shown in the indicated figure.

Thank you for catching this. We fixed the error.

4. Supp Fig S6 needs a visible control.

We adjusted the control image to make the background visible.

5. In some cases (e.g., Fig 2a) axis text on plots is not readable.

Figure 2 is now the revised Figure S4. Following the suggestion, the size of the fonts has been increased.

Reviewer #5 (Remarks to the Author):

Point-by-point response to reviewer comments

We thank Reviewers #1, #2, #4, and #5 for accepting our revisions. Below is a summary of our response to the additional remarks by Reviewer #3.

Reviewer #3

The authors did a great job and the manuscript improved considerably. Thank you for that. However, I still have a few issues that, IMHO, must be resolved before the manuscript can be considered suitable for publication in this journal.

1) Figs 1j, 5g, 7n, etc. As I noted in my original comment, to use tubulin as a normalization control for the anti-P-eIF2 signal is technically very wrong and any conclusion driven from these experiments concerning the eIF2-independence of the observed effects is dangerously misleading (e.g. lines 347-348; but mainly the closing paragraph 433 – 439!). Moreover, taking into account that the newly added proteomics revealed the RNAi analysis of 4EHP and NELF-E increased levels of all three eIF2 subunits! To examine if the P-status of eIF2-alpha has changed or not under given stress/condition, the anti-P-eIF2 signal has to be normalized only to anti-eIF2alpha (in the worst case to other eIF2 subunits, although they do not always occur in the equimolar ratio). Please check the old scholarly papers by Hinnebusch, Wek, Ron, Sonenberg... It really is a must.

To avoid making any misleading conclusions about eIF2-independence, we deleted all references to “eIF2-independent” regulation in this revised version. This includes lines 347-348 and the closing paragraph. The phospho-eIF2 data was peripheral to our primary focus on 40S and eIF3, and therefore, we moved the eIF2 α blots to Supplemental Figures S6, S11, and S17. In those Figures, the total eIF2 α blots corroborate the proteomic data indicating increases in total eIF2 α . Those Figures also include phospho-eIF2 α blots and new graphs showing phospho-eIF2 α normalized to total eIF2 α .

2) I understand that the human SL3 might not be present everywhere. We have seen it ourselves. What my comment was supposed to imply is the following (and I really do not care if the authors cite our most recent paper describing SL3 or not – it is not about this!). Ignoring this uORF2/ATF4 overlap region in the past generated confusing/conflicting results. One way or the other, it is a conserved region, because it is a coding region. Therefore, it is likely, maybe even highly likely, that it contains some mRNA features acting in-trans even in D.m. Have you checked its potential to form stable sec. structures? Your construct in Fig. 7 is responding as expected, but can you guarantee that you are monitoring the physiological response? Perhaps this overlap makes it weaker or stronger? Maybe not, we do not know, but why to repeat the same “mistake”? If anything, the authors should state clearly in the main text that the fact that their construct lacks this region represents a possible limitation to its ability to respond at the full physiological scale. And again, is it perfectly fine to ignore us :)

We do not argue that our reporter monitors ALL physiological response. Using a reporter that responds a limited aspect of crc (ATF4) regulation has its utility, because it can provide specific mechanistic insights. Following the reviewer suggestion, we now add a line in the main text stating that “The transgene will not capture potential crc (ATF4) regulation that acts through the protein-coding sequence but is designed to report regulatory inputs at the 5' leader upstream of the main ORF” (lines 340 – 342).

This thinking brought me back to the title. Considering these and other differences, I guess the title should make a reader aware of the fact that this work explored D.m.; e.g. “4EHP and NELF-E regulate physiological 1 ATF4 induction and proteostasis in disease models of *Drosophila melanogaster*.” Or something like this.

Following the suggestion, we changed the title.

3) A few notes to consider:

- there is literature showing that eIF3l and 3k are considered to be inhibitory (not initiation-stimulatory) subunits

- lines 301 – 303; what do you mean by “a compensatory mechanism”? Specific reduction in biogenesis of one ribosomal subunit will naturally lead to an increase of the other subunit.

To clarify, we revised the sentence to “feedback regulatory mechanism against RpS reduction” (lines 307, 308).

- 303 – 307; this is really interesting; It seems that the 40S bio defect lies in an impaired formation of the 40S beak. Can you check in Ramakrishnan’s Science paper (2020) where exactly the positions of 3l and 3k in the 48S PIC were mapped? They can exist as a dimer out of eIF3 and thereby they might be needed for final maturation of the 40S, which would explain a lot.

The below image shows our Figure 6f (left) juxtaposed with the 48S structure reported by Ramakrishnan and colleagues (Science 2020, PMID 32882864). The orientation of the 40S is similar in both images, with the beak region at the upper right corner. RpS10, RpS27A, and RpS12 constitute the beak region the 40S structure (left). On the other hand, eIF3l and eIF3k bind the opposite end of the 40S (right).

Also, there is an EMBO J. paper from D. Wolf's lab (2022?) on eIF3k and ribosomal proteins with respect to 40S biogenesis that could help too.

Thank you for letting us know. We will consider this in our next study when we examine the relationship between eIF3 and ribosome biogenesis (which is not the main subject of the current work).

- lines 310 – 314; If my assumption is correct, I would be very curious to see if the levels of RPS go down and those of RPL go up in RPS12-S2383/+ in a similar fashion as observed with 4EHP and NELF-E siRNAs. This could nail the mol mech down.

In page 9, we had cited other reports that “reduction of one RpS subunit causes the reduction of other RpS subunit proteins while increasing RpL subunits (ref. 44, 45)” (lines 186, 187). Refs 44 and 45 had examined RpS3 +/- (ref. 44) and RpS23 +/- . Both proteomic studies show reduction of other RpS and a concomitant increase in most RpLs. So, the reviewer's assumption is supported by published data.

- 318; I would be more accurate: “the post-termination 40S subunits”.

Thank you. We revised it accordingly in line 322.

- 321; eIF3j is not a bona fide eIF3 subunit, it is only an eIF3-associated factor

Thank you. We changed the sentence to “RNAi lines targeting eIF3 subunits eIF3h and eIF3l, and the eIF3-associated factor eIF3j, did not interfere with larval development” (line 325).

- 419; for the sake of collegiality, please also include 26 and 29; this is important to us.

The references have been added.

- 425 – 426; for same reason, it would be fair to add that a similar observation was made in humans (31).

The reference has been added.